# Massively parallel reporter assays of melanoma risk variants identify *MX2* as a gene promoting melanoma

Jiyeon Choi [1,10], Tongwu Zhang [1,10], Andrew Vu[1], Julien Ablain [2], Matthew M. Makowski[3], Leandro M. Colli[1], Mai Xu[1], Rebecca C. Hennessey[1], Jinhu Yin[1], Harriet Rothschild[2], Cathrin Gräwe[3], Michael A. Kovacs[1], Karen M. Funderburk[1], Myriam Brossard [4], John Taylor [5], Bogdan Pasaniuc[6], Raj Chari[7], Stephen J. Chanock [1], Clive J. Hoggart[8], Florence Demenais [4], Jennifer H. Barrett [5], Matthew H. Law [9], Mark M. Iles [5], Kai Yu[1], Michiel Vermeulen [3], Leonard I. Zon [2] & Kevin M. Brown [1✉]

Genome-wide association studies (GWAS) have identified ~20 melanoma susceptibility loci, most of which are not functionally characterized. Here we report an approach integrating massively-parallel reporter assays (MPRA) with cell-type-specific epigenome and expression quantitative trait loci (eQTL) to identify susceptibility genes/variants from multiple GWAS loci. From 832 high-LD variants, we identify 39 candidate functional variants from 14 loci displaying allelic transcriptional activity, a subset of which corroborates four colocalizing melanocyte *cis*-eQTL genes. Among these, we further characterize the locus encompassing the HIV-1 restriction gene, *MX2* (Chr21q22.3), and validate a functional intronic variant, rs398206. rs398206 mediates the binding of the transcription factor, YY1, to increase *MX2* levels, consistent with the *cis*-eQTL of *MX2* in primary human melanocytes. Melanocyte-specific expression of human *MX2* in a zebrafish model demonstrates accelerated melanoma formation in a *BRAF*^V600E background. Our integrative approach streamlines GWAS follow-up studies and highlights a pleiotropic function of *MX2* in melanoma susceptibility.

[1] Division of Cancer Epidemiology and Genetics, National Cancer Institute, Bethesda, MD 20892, USA. [2] Stem Cell Program and Division of Hematology/Oncology, Boston Children's Hospital and Dana-Farber Cancer Institute, Boston, MA 02115, USA. [3] Department of Molecular Biology, Faculty of Science, Radboud Institute for Molecular Life Sciences, Oncode Institute, Radboud University Nijmegen, 6525 XZ Nijmegen, The Netherlands. [4] Université de Paris, UMRS-1124, Institut National de la Santé et de la Recherche Médicale (INSERM), F-75006 Paris, France. [5] Leeds Institute for Data Analytics, School of Medicine, University of Leeds, Leeds LS2 9JT, UK. [6] Department of Human Genetics, David Geffen School of Medicine, University of California, Los Angeles, Los Angeles, CA 90024, USA. [7] Genome Modification Core, Frederick National Lab for Cancer Research, National Cancer Institute, Frederick, MD 21701, USA. [8] Department of Medicine, Imperial College London, London SW7 2BU, UK. [9] Statistical Genetics, QIMR Berghofer Medical Research Institute, Brisbane, QLD 4006, Australia. [10] These authors contributed equally: Jiyeon Choi, Tongwu Zhang. ✉email: kevin.brown3@nih.gov

A series of genome-wide association studies (GWAS) over the past decade has identified about twenty genomic loci associated with cutaneous melanoma[1–10], highlighting the genetic contribution to melanoma susceptibility in the general population. Some of these loci represent genes or regions implicated in melanoma-associated traits e.g., pigmentation phenotypes[11–15] and nevus count[5,16,17]. Other than these loci, however, underlying mechanisms of genetic susceptibility to melanoma in the general population is less well understood. For a small number of these loci, extensive characterization of susceptibility genes and variants under the GWAS peaks have led to insights into molecular pathways underlying melanoma susceptibility. *PARP1*, located in the Chr1q42.1 melanoma locus[8], was shown to be a susceptibility gene that has tumor-promoting roles in early events of melanomagenesis through its regulation of melanocyte master transcription factor and oncogene, *MITF*[18], while a functional variant at a multi-cancer locus on Chr5p15.33 was characterized highlighting the role of *TERT* in cancer susceptibility including in melanoma[19]. Still, the molecular mechanisms underlying the majority of common melanoma risk loci remain unexplained.

Recent advances in sequencing technologies have enabled a number of classical molecular assays to be conducted at a large scale. Massively Parallel Reporter Assays (MPRA) scale up conventional luciferase reporter assays for testing transcriptional activities of DNA elements, facilitating evaluation of tens of thousands of different short sequences at the same time in cells, which are then deconvoluted by massively parallel sequencing[20–22]. Incorporation of this approach is particularly attractive for GWAS functional follow-up studies, as (1) linkage disequilibrium (LD) limits statistical fine-mapping and leaves numerous variants as potential functional candidates, and (2) many trait-associated variants are hypothesized to contribute to allelic gene expression through *cis*-regulatory mechanisms that can be tested by reporter assays. Therefore, direct assessment of allelic differences in transcriptional regulation could help prioritize likely functional variants among multiple variants tied by LD. For example, a recent study adopted MPRA to test 2756 variants from 75 GWAS loci for red blood cell traits and identified 32 functional variants from 23 loci[20].

In addition, expression quantitative trait loci (eQTL) analysis can be a powerful approach for identifying susceptibility genes from GWAS loci, as it informs on genes for which expression levels are correlated with trait-associated variants. While there are a number of publicly available eQTL datasets using tissues representing different human organs including those through the GTEx project[23], most of them are based on bulk tissue samples (e.g., skin tissues) as opposed to individual cell types. Importantly, melanomas arise from melanocytes, but they account for less than 5% of a typical skin biopsy. To dissect cell-type specific gene expression regulation implicated in melanoma predisposition, a melanocyte eQTL dataset using primary cultures of melanocytes from 106 individuals was established and mapped six melanoma GWAS loci (30% of all the loci) to melanocyte eQTLs[24]. This dataset identified more candidate susceptibility genes than using eQTLs from datasets of larger sample size generated from bulk skin tissues, other tissue types from GTEx, and melanoma tumors[24], highlighting the utility of cell-type specific eQTL dataset for functional follow-up of GWAS regions.

In this study, we combine MPRA and cell-type specific melanocyte eQTL to scale up the functional annotation process for melanoma GWAS loci and nominate the best candidates for testing in a zebrafish model. Using our approach we identify a functional risk variant that increases the level of an HIV-1 restriction gene, *MX2*, in cells of melanocytic lineage; subsequent expression of *MX2* in melanocytes of a zebrafish melanoma model accelerates melanoma formation.

## Results

**MPRA identified melanoma-associated functional variants.** To identify functional melanoma-associated variants displaying allelic transcriptional function, we used the MPRA approach. Among 20 genome-wide significant melanoma loci from the most recent GWAS meta-analysis[1], we prioritized 16 loci where a potential *cis*-regulatory mechanism could be hypothesized, excluding four pigmentation-associated loci previously explained by functional protein coding variants (*MC1R*, *SLC45A2*, and *TYR*[11–14]) or shown not to be expressed in melanocytes (*ASIP*[15]). To comprehensively analyze genetic signals from these loci, we then performed statistical fine-mapping using the HyperLasso[25] approach. The fine-mapping nominated additional independent signals (Supplementary Table 1), from which we selected 30 variants, adding to the 16 lead SNPs from the initial meta-analysis results[1]. To prioritize melanoma-associated variants to test by MPRA, we first selected 2748 variants that are in LD ($r^2 > 0.4$) with these 46 primary and secondary lead SNPs (Methods; Supplementary Fig. 1; Supplementary Table 2). Among them, we further prioritized 832 variants that overlap potentially functional melanoma-relevant genomic signatures, namely, open chromatin regions and promoter/enhancer histone marks in primary melanocytes and/or melanoma short term cultures[26] (Supplementary Data 1; Supplementary Table 3; Methods; www.encodeproject. org; www.roadmapepigenomics.org). We then constructed MPRA libraries for these 832 variants using methods adopted from previous studies[20–22,27]. A 145 bp genomic sequence encompassing the risk or protective allele of each variant was tested for their potential as an enhancer or promoter element in luciferase constructs. For each variant, a scrambled sequence for its core 21 bases was also tested as a null (Fig. 1a; Methods). Transcribed output of tag (barcode) sequences associated with each tested DNA element were then measured by sequencing, after transfections into a melanoma cell line (UACC903) to represent melanoma-specific *trans*-acting factors and the HEK293FT cell line to obtain maximum transfection efficiency. From these data, we initially observed significantly high correlation of transcriptional activities among replicates, and further applied a conservative quality control measure for downstream analyses (Methods; Supplementary Figs. 2–6; Supplementary Table 4).

To nominate variants displaying allelic transcriptional activity, we focused on those displaying significant difference between two alleles (FDR < 0.01; two-sided Wald test with robust sandwich type variance estimate; multiple comparisons adjusted using Benjamini & Hochberg method), and then further selected those with either allele displaying a significant departure from the null (scrambled core sequence; FDR < 0.01) (Supplementary Fig. 2). After applying these cutoffs, 39 of the 832 tested variants (~4.7%) qualified as displaying allelic transcriptional activity in the UACC903 melanoma cell dataset alone, as well as in the combined total dataset (Methods; Supplementary Fig. 7a; Supplementary Table 5). These candidate functional variants are from 14 melanoma GWAS loci with 1–9 variants per locus (median 1.5 variants), demonstrating that MPRA narrowed down functional candidate variants to a considerably small number from tens to hundreds of high-LD variants in most of the loci (Fig. 1b; Supplementary Table 6; Supplementary Fig. 8). Transcriptional activities of these 39 variants were significantly higher than those of negative controls (8 variants of high LD with the lead SNP but located in non-DHS/non-promoter/enhancer histone mark in melanocytes/melanoma cells; $P < 2.2e{-}16$, effect size = 0.137; Mann–Whitney $U$ test; Supplementary Fig. 7b), as well as the rest of the variants (non-significant variants; $P < 2.2e{-}16$, effect size = 0.109). These 39 variants displayed a modest 1.13 to 3.49-fold (median 1.26-fold) difference in transcriptional activity between two alleles consistent with subtle

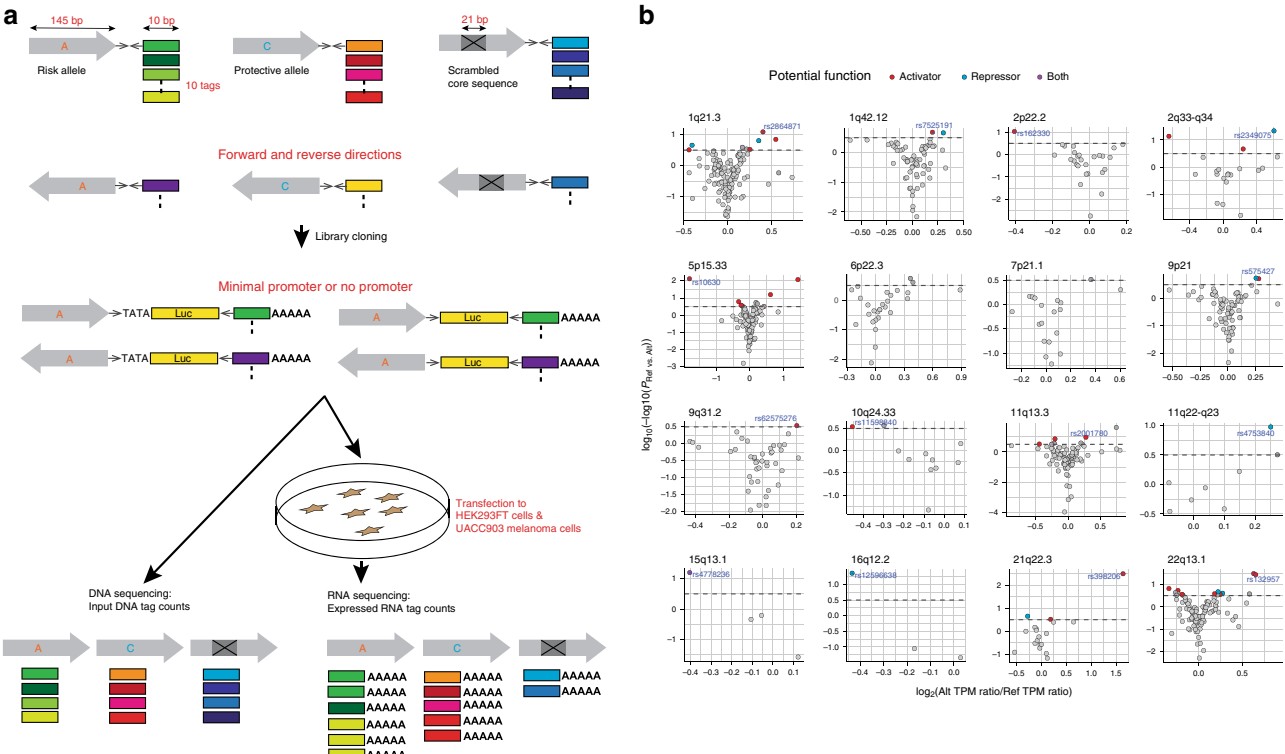

**Fig. 1 MPRA identified 39 functional variants from 16 melanoma GWAS loci. a** MPRA workflow. Oligo libraries were synthesized using 145 bp of sequence encompassing each variant with risk or protective alleles or a scrambled sequence for core 21 bases in both forward and reverse directions, that was flanked by enzyme recognition sites and sequencing primer sequences, as well as 10 bp barcodes (10 tags per unique sequence). Libraries were cloned into luciferase constructs with or without a minimal TATA promoter. Cloned libraries were then transfected into HEK293FT cells or UACC903 melanoma cells to generate expressed RNA tag libraries. Both input DNA and RNA libraries were sequenced to assess the tag counts associated with the test sequences. Luc: luciferase gene, AAAAA: poly-A tail. **b** Volcano plots of MPRA results for each melanoma GWAS locus. Inverse P-values and effect sizes of allelic difference from UACC903 transfections are shown for each of the 16 loci tested. A two-sided Wald test with robust sandwich type variance estimate was used. Multiple comparisons were adjusted using the Benjamini and Hochberg method. Dashed horizontal lines indicate the FDR 1% cutoff for allelic difference in the UACC903 set. The most significant variant from each locus is labeled. Putative function of 39 significant MPRA variants are shown as activator (red circle), repressor (blue circle), or both (purple circle) (expression levels of either allele is higher, lower, or higher and lower than those of scrambled sequence, respectively). Gray variants above the FDR 1% cutoff are those that failed additional criteria (allelic difference in the combined data or significant departure from the scrambled control). No significant variants were identified from the loci on Chr6p22.3 and Chr7p21.1. Source data are provided as a Source Data file.

effects on risk observed for common cancer-associated variants (UACC903 cells; Supplementary Table 5). We then asked if the observed allelic differences from MPRA are in part due to differential binding of transcription factors. For this, we predicted allelic transcription factor binding affinity of each tested variant using motifbreakR[28] and subsequently correlated the predicted allelic binding scores with the allelic transcriptional activities measured from MPRA. Notably, the MPRA-significant variants displayed a higher level of correlation compared to that of non-significant ones, shown by a larger Pearson $r$ (Pearson $r = 0.24$ vs. $-0.023$; Supplementary Fig. 9a), while the P-values for the both tests are not significant ($P = 0.149$ and $0.556$, respectively). We then asked if MPRA-significant variants are enriched in credible causal variants or those with higher probability scores from a Bayesian fine-mapping approach. For this we performed statistical fine-mapping of melanoma GWAS data using PAINTOR[29] while integrating primary melanocyte-specific functional annotations (Methods). When overlaid with the fine-mapping results, the 39 significant MPRA variants (FDR < 0.01) indeed displayed the highest median probability score compared to other variant groups with varying FDR cutoffs, which was a 2.12-fold enrichment over all the tested variants with probability scores (Supplementary Fig. 9b). These data demonstrated that MPRA can quickly narrow down to a small number of plausible

functional candidate variants from tens to hundreds of high-LD variants from melanoma GWAS loci by measuring allelic transcriptional activity.

**eQTL prioritized functional candidates in four melanoma loci.** To prioritize functional variants that contribute to melanoma risk through regulation of nearby gene expression, we turned to cell-type specific melanocyte eQTL data from 106 individuals[24]. 597,335 significant *cis*-eQTL SNPs (+/−1 Mb of TSS, FDR < 0.05, not LD-pruned) were identified in this dataset, with 6 of 20 melanoma GWAS loci displaying significant co-localization/TWAS[24]. For the purpose of nominating the most plausible candidates for functional follow-up, we mainly focused on the five of these six loci with melanocyte eQTL support (1q21.3, 1q42.12, 2q33-q34, 21q22.3, and 22q13.1) that were also tested in our MPRA. While there were initially 17 to 116 eQTL variants per locus at a genome-wide significant level, MPRA further narrowed them down to 3 to 9 functional variants in four loci based on transcriptional activity (Supplementary Table 6). Among them, a total of nine variants from four loci displayed a consistent direction between MPRA and eQTL, in which the direction of allelic expression of local genes matches those of MPRA allelic transcriptional activity (Supplementary Table 6; Supplementary

Fig. 9c). Namely, two MPRA-significant variants (rs2864871 and rs6700022) from the locus on chromosome band 1q21.3 were significant eQTLs for *CTSS* in melanocytes, where lower *CTSS* levels were correlated with melanoma risk. Similarly, two to three variants each (rs2349075, rs529458487, rs398206, rs408825, rs4383, rs4384, and rs6001033) from three other loci (2q33-q34, 21q22.3, and 22q13.1) also overlapped with melanocyte eQTLs, where lower *CASP8*, higher *MX2*, and higher *MAFF* levels were correlated with melanoma risk, respectively (Supplementary Table 7). For 1q42.12, none of the eQTL significant variants passed the MPRA significance cutoff, and therefore were not prioritized. Thus, by combining MPRA and cell-type specific melanocyte eQTL, we efficiently prioritized candidate functional variants and susceptibility genes from multiple melanoma GWAS loci.

**MX2 is a susceptibility gene in the 21q22.3 melanoma locus**. To validate the efficiency of our prioritization scheme, we performed a thorough validation and functional characterization for one of the four loci with combined MPRA and melanocyte eQTL support. We chose the locus on 21q22.3, where MPRA identified the most significant variant. In this locus, twenty-two variants were originally tested in MPRA (Supplementary Table 8), nineteen of which are located in the first intron of the *MX2* gene (Fig. 2a), and the remaining three upstream. Of these, three variants were significant in MPRA, and rs398206, in particular, (Fig. 2a, shown in magenta) displayed the lowest P-value of all 832 tested variants. rs398206 displayed a strong transcriptional activator function (1.7 to 4.3-fold above the scrambled sequence), as well as significant allelic difference, where the melanoma risk-associated A allele drove significantly higher luciferase expression than protective C allele (3.1-fold in UACC903 cells, FDR = 5.6e−206; Fig. 2b). Subsequent individual luciferase assays using the same 145 bp sequence in two melanoma cell lines validated this finding (2.7 to 5.0-fold allelic difference, *P* = 1.1e−6–5.2e−11; two-tailed, unpaired *t*-test assuming unequal variance; Fig. 2c; Supplementary Fig. 9d). rs398206 was also a significant eQTL for levels of *MX2* gene expression in primary melanocytes, where the melanoma risk-associated A allele is correlated with higher *MX2* expression (Slope = 0.70, *P* = 6.6e−15; linear regression; Fig. 2d).

While melanocyte eQTL consistently identified *MX2* as the best candidate susceptibility gene at the 21q22.3 melanoma locus[24], we further interrogated eQTL data from melanocytes and 44 GTEx tissue types, to comprehensively assess potential melanoma susceptibility gene(s) in this locus. When we inspected eQTL data from 44 GTEx tissue types, rs398206 was a significant eQTL for *MX2* in five other tissue types (testis, transformed skin fibroblasts, ovary, tibial nerve, and whole blood) but no other gene displayed a genome-wide significant eQTL with rs398206 (using a nominal P-value threshold set for each gene based on a genome-wide empirical P-value as defined by GTEx V6p; GTEx portal; https://gtexportal.org).

As the melanocyte *cis*-eQTL analyses used for the above assessments were limited to the genes in +/−1Mb of the tested variants[24], we explored if rs398206 is a marginal eQTL for any gene in the topologically-associated domain (TAD) to account for potential gene regulation mediated by chromatin looping typically occurring within this physical domain. From the genomic interval defined as the TAD encompassing rs398206 (chr21:42,480,000–44,320,000; hg19; retrieved from Hi-C data of SKMEL5 melanoma cell line generated for ENCODE dataset via http://promoter.bx.psu.edu/hi-c/), a total of 21 genes were significantly expressed in melanocytes, for which eQTL analyses were performed. The results demonstrated that *MX2* displayed the most significant eQTL with rs398206 (*P* = 6.6e−15; linear

regression), while none of the other genes in the TAD displayed even a marginally significant eQTL after adjusting for multiple testing (Bonferroni-corrected cutoff at *P* < 0.0024 for 21 genes; Supplementary Table 9). These data determined that *MX2* is the most likely susceptibility gene at the 21q22.3 melanoma susceptibility locus.

To complement the eQTL data, we also assessed allele-specific expression (ASE) of *MX2* in melanocytes. rs398206 is located in the 5′ UTR region of an alternative *MX2* transcript isoform (ENST00000543692; Supplementary Fig. 10a), the expression levels of which are correlated with the most abundant full-length transcript in melanocytes (ENST00000330714; Pearson *r* = 0.69, *P* = 1.63e−16; Supplementary Fig. 11). RNA sequencing data from our previous study did not find genome-wide significant ASE for any melanoma-associated SNP (GWAS *P* < 5e−8) residing in the transcribed region of *MX2*[24], partly due to low sequence coverage of this transcript that is expressed at a low level. To thoroughly examine allele-specific expression in this region, we genotyped rs398206 in melanocyte cDNA using a Taqman genotyping assay that recognizes both genomic DNA and cDNA. The results demonstrated an over-representation of A allele-bearing transcripts in 27 heterozygous individuals, when the allelic ratio in cDNA was normalized to those in genomic DNA (One-sample Wilcoxon test, *P* = 2.49e−5; Supplementary Fig. 12). These data are consistent with the eQTL data, where the risk-associated A allele is correlated with higher *MX2* expression.

To thoroughly investigate possible mechanisms of allelic *MX2* expression in relation to rs398206, we performed additional QTL analyses in melanocytes addressing alternative modes of gene regulation, including splice-QTL (sQTL) and DNA methylation QTL (meQTL). sQTL analyses using LeafCutter[30] suggested that the main effect of the *MX2* eQTL was not driven by alternative isoforms or splicing events (Supplementary Fig. 10b–f; Supplementary Note). meQTL analysis, on the other hand, identified a significant meQTL for rs398026 at a CpG probe near the *MX2* canonical promoter, where the melanoma risk-associated A allele is correlated with lower CpG methylation, which is consistent with higher expression of the full-length isoform (Supplementary Fig. 13). Two other CpG probes in the first intron of *MX2* (closer to rs398206) also displayed significant meQTLs for rs398206 in melanocytes, where higher CpG methylation is correlated with the risk A allele. These observations are consistent with the previous findings that DNA methylation in promoters is negatively correlated with gene expression, while that of transcribed regions is positively correlated with gene expression[31–35]. Taken together, eQTL, sQTL, and meQTL data are consistent with the hypothesis that *MX2* full-length transcript mainly accounts for the eQTL at rs398206 in melanocytes through a transcriptional mechanism.

**rs398206 regulates MX2 levels via allelic binding of YY1**. To identify protein factors mediating the allelic difference observed in MPRA, we performed comparative mass-spectrometry using a 21 bp DNA probe encompassing rs398206 with A or C alleles and nuclear extract from the UACC903 melanoma cell line (Fig. 3a). Among the proteins displaying allelic binding, the most prominent A-allele preferential binding was shown for Yinyang-1 (YY1), a ubiquitous transcription factor having roles in development and cancer[36], as well as in pigmentation pathways of melanocytes[37]. Sequence-based motif prediction was also consistent with this finding, indicating that the sequence around rs398206 forms a consensus binding site for YY1 favoring the A-allele (Fig. 3b). Subsequent electrophoretic mobility shift assays (EMSAs) validated that this A-allele-preferential binding of nuclear proteins is sequence-specific, as shown by competition

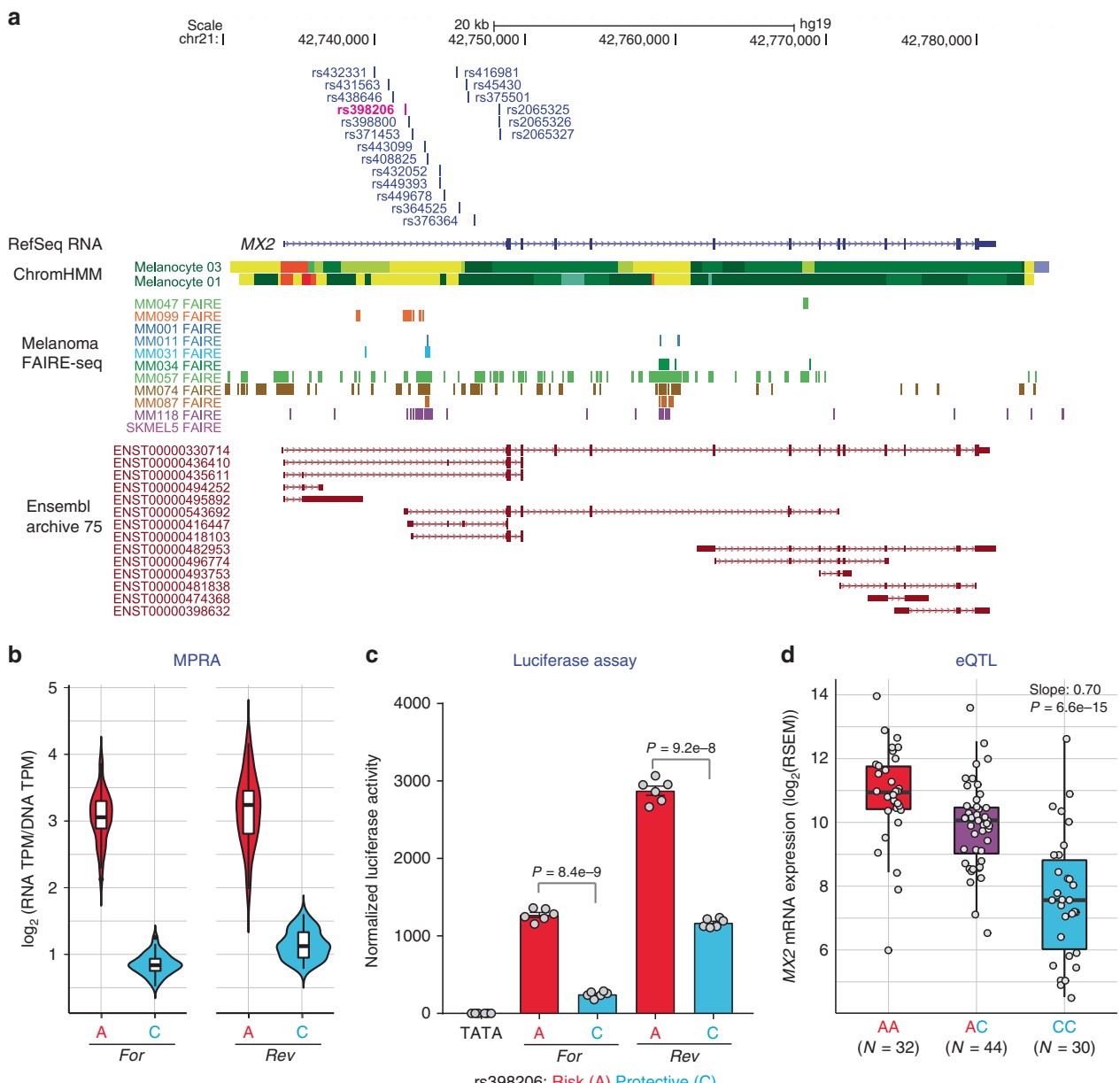

**Fig. 2 rs398206 is a functional *cis*-regulatory variant and a significant *cis*-eQTL for *MX2* levels in melanocytes. a** Variants that were tested in MPRA from the Chr21q22.3 melanoma locus are shown relative to the genomic position of *MX2*. Only the 19 variants located in the first intron of *MX2* coming from the primary GWAS signal are shown (the other three from a secondary signal are located upstream of the *MX2* genic region). ChromHMM annotation (Primary Core Marks segmentation) of Penis Foreskin Melanocyte Primary Cells from Roadmap Epigenomics Project is shown (Red/OrangeRed: Active_TSS/Flanking_Active_TSS, Yellow/GreenYellow: Enhancers/Genic_enhancers, Green/DarkGreen: Strong_transcription/Weak_transcription). Melanoma FAIRE-seq track of 11 samples is from a study by Verfaillie and colleagues[26]. Ensembl predicted transcripts from archive 75 are shown. **b** Transcriptional activity of 145 bp sequences encompassing rs398206 from MPRA are shown as normalized tag counts ($\log_2$(RNA TPM/DNA TPM)). Results from UACC903 melanoma cells are shown for both alleles in forward (For) and reverse (Rev) directions, where results from promoter and enhancer constructs were combined. Center lines show the medians; box limits indicate the 25th and 75th percentiles as determined by R software; whiskers extend 1.5 times the interquartile range from the 25th and 75th percentiles, outliers are represented by dots. Density is reflected in the width of the shape. **c** Individual luciferase activity assays of 145 bp sequences encompassing rs398206 is shown for UACC903. pGL4.23 construct including minimal TATA promoter was used. One representative set is shown from three biological replicates. Mean with SEM, $n = 6$. All constructs are significantly higher than pGL4.23 (TATA) control ($P < 0.0001$, two-tailed, unpaired t-test assuming unequal variance). **d** eQTL plot of *MX2* levels in primary melanocytes in relation to rs398206 is shown for three genotype groups. Center lines show the medians; box limits indicate the 25th and 75th percentiles; whiskers extend 1.5 times the interquartile range from the 25th and 75th percentiles. *P*-value and slope were derived from linear regression with no multiple-testing correction applied. Source data are provided as a Source Data file.

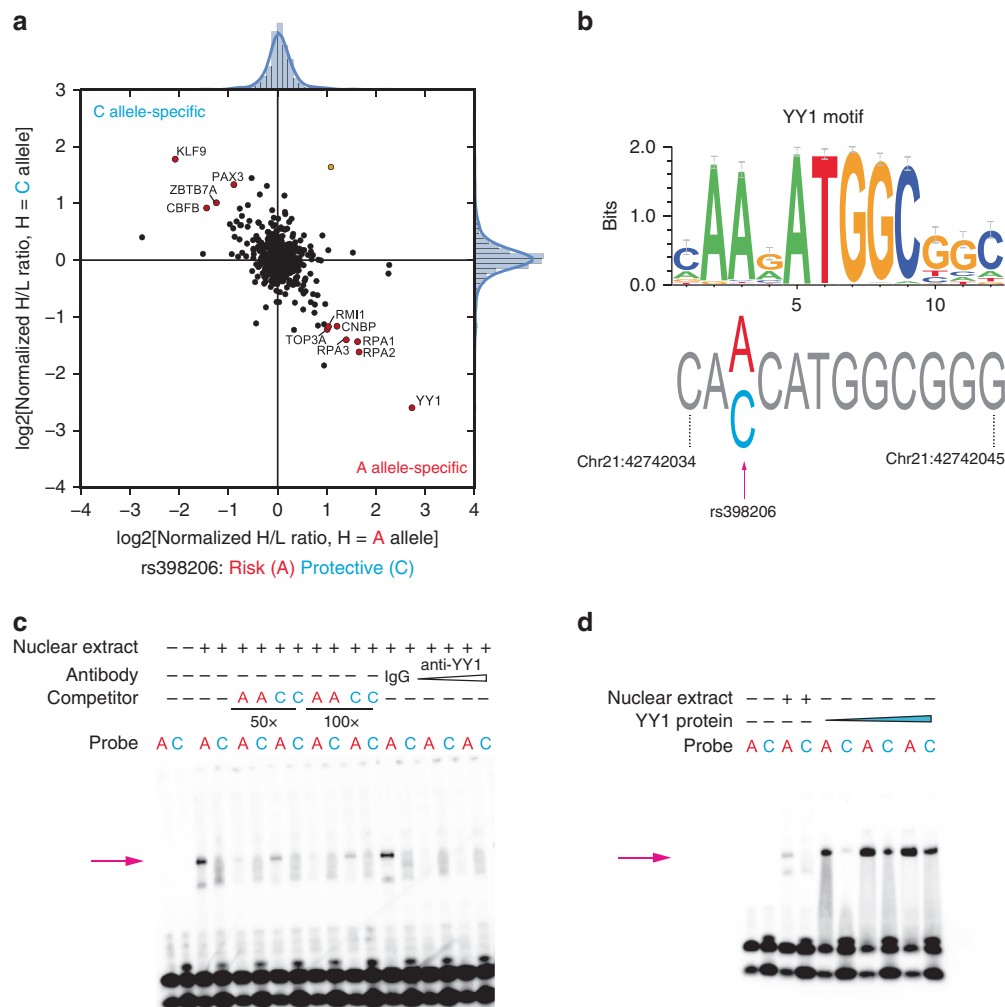

**Fig. 3 rs398206 displays allele-preferential binding to YY1. a** Quantitative mass-spectrometry of rs398206 using nuclear extract of UACC903 and 21 bp double-stranded DNA probes with A (risk) or C (protective) alleles. A-allele specific interacting proteins are shown in the bottom right quadrant, and C-allele specific interactors in the top left quadrant. Using label-swapping of high(H)-mass or low(L)-mass label, the A-bound/C-bound ratio is shown on the x-axis, and the C-bound/A-bound ratio on the y-axis. Proteins passing the inter-quartile >3 cutoff for both axes are color-coded for consistent (red circle) and inconsistent (yellow circle) direction from label-swapping. Names of consistently identified proteins are shown. **b** YY1 binding motif is shown as a position weight matrix at the top (motif obtained from HOCOMOCO database and plotted using weblogo3). The genomic sequence surrounding rs398206 is shown at the bottom with the risk-associated A allele matching the consensus YY1-binding motif. Genomic positions are hg19. **c–d** EMSAs using 21 bp double-stranded DNA probes with **a** or **c** alleles of rs398206 and nuclear extract from UACC2331 melanoma cells (**c**) or purified recombinant YY1 protein (**d**). Antibody super-shift using anti-YY1 antibody is shown at the last lanes of **c**, where the A-specific band (arrows) is diminished. Representative sets from three replicates (**c**) and one replicate (**d**) are shown. Source data are provided as a Source Data file.

with unlabeled probes (Fig. 3c). Antibody super-shift demonstrated that YY1 is present in this subset of allelic-binding proteins (Fig. 3c), which was further validated by EMSAs with purified recombinant YY1 protein (Fig. 3c, d). We subsequently performed chromatin immunoprecipitation (ChIP) using anti-YY1 antibody and scanned a ~5 Mb genomic region encompassing rs398206 in three melanoma cell lines representing each genotype of rs398206 (AA, AC, and CC). We observed prominent enrichment of YY1 binding on top of rs398206 in the AA cell line, a weaker but clear enrichment in the AC line, and even weaker binding enrichment over rs398206 in the CC line (Fig. 4a; Supplementary Fig. 14). Given that differences between cell lines (e.g., DNA copy number differences, accessibility of chromatin in the region encompassing rs398206, YY1 levels, and variability in formaldehyde fixing and chromatin shearing efficiency) may also contribute to differential YY1 binding, we also assessed allele-specific YY1 binding in the heterozygous AC cell line

(UACC647). We performed genotyping of rs398206 using the DNA fragments pulled down by anti-YY1 antibody. DNA fragments pulled down using YY1 antibody displayed a significant enrichment of A allele (Mann-Whitney $U$ test, $P = 9.1e{-}3$), while genomic DNA and serial-diluted input DNA displayed equivalent signal from both A and C alleles, indicating clear A-allele preferential binding of YY1 in melanoma cells (Fig. 4b, c).

Based on this strong allelic YY1 binding, we next asked if YY1 regulates endogenous *MX2* expression levels. siRNA knockdown of YY1 in the UACC903 melanoma cell line demonstrated a weak but consistent reduction of *MX2* levels by four different sets of siRNAs (14–32% decrease, $P = 1.5e{-}3{-}1.9e{-}5$, one-sample Wilcoxon test; Fig. 5a; Supplementary Fig. 15a) indicating a regulation of *MX2* levels by YY1. To further determine if the genomic region encompassing rs398206 regulates endogenous *MX2* levels, we targeted this region by CRISPRi using dCAS9-KRAB-MeCP2[38] in the same melanoma cell line. Four gRNAs

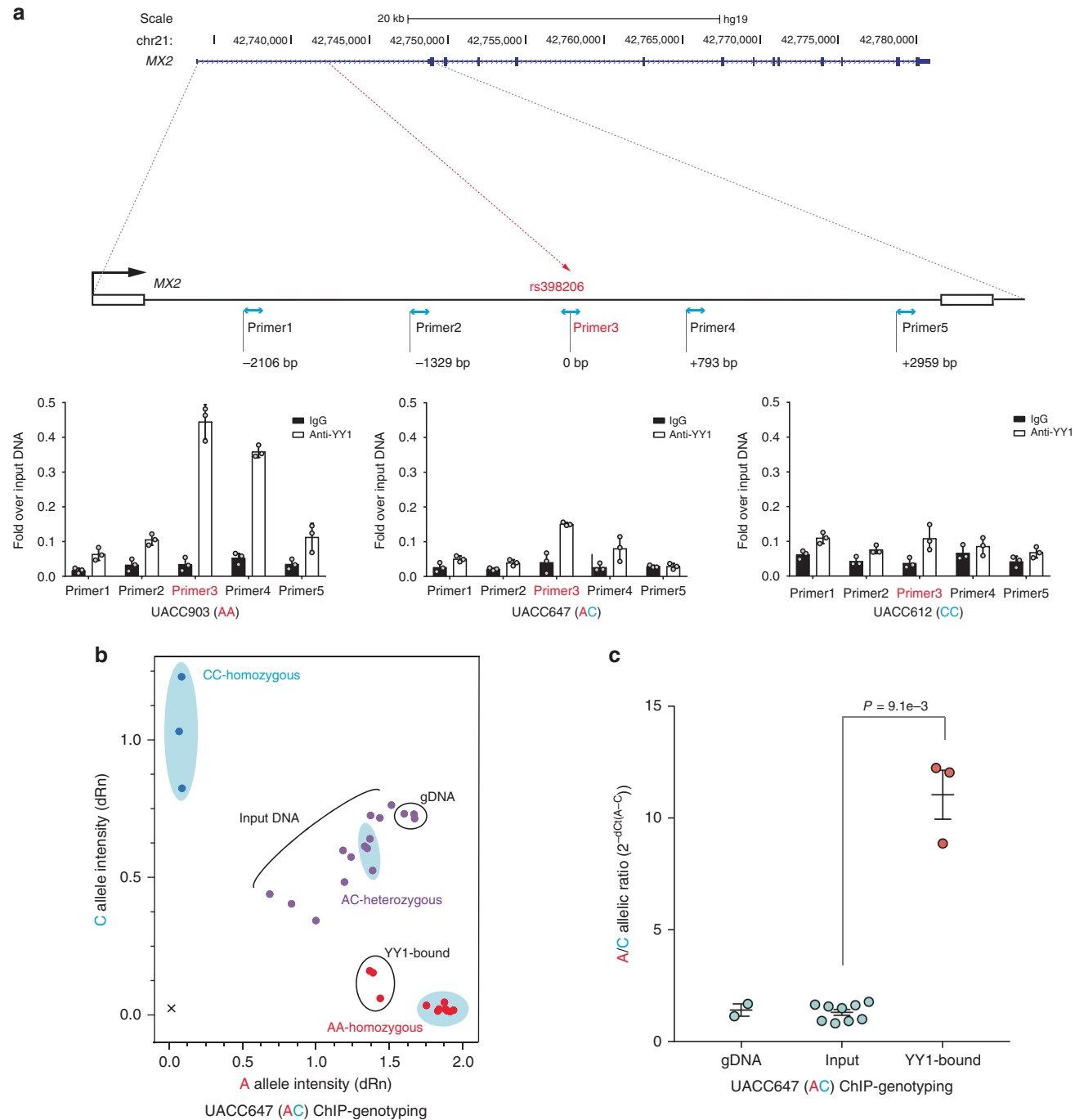

**Fig. 4 YY1 preferentially binds to the rs398206-A allele in melanoma cells. a** Chromatin immunoprecipitation using anti-YY1 antibody or normal IgG followed by qPCR. On the top panel, genomic positions of the amplicons using five qPCR primer sets are shown relative to rs398206 (red dashed line) in the first intron of *MX2*. qPCR results from three melanoma cell lines are shown on the bottom panel with genotype for rs398206 indicated in the *X*-axis label. Relative quantities are shown as fold over input DNA. Mean of PCR triplicates with SD are plotted. One representative set of three biological replicates for each cell line is shown. **b** Taqman genotyping of rs398206 using YY1 ChIP DNA in UACC647 melanoma cells (heterozygous for rs398206). The blue shaded areas mark HapMap CEU DNA controls showing separation of three genotype clusters. qPCR triplicates were plotted separately. One representative set from three biological replicates is shown. Normalized A and C allele intensity is shown as dRn values on *x* and *y*-axis, respectively. gDNA: genomic DNA. **c** Combined plot for A/C allelic ratio from three rounds of biological replicates of UACC647. Each dot represents A/C ratio calculated from $2^{-(average\ dCt(alleleA-alleleC))}$ from qPCR triplicates. Input includes three 10-fold serial dilutions from each round. Mean with SEM are plotted. A two-tailed Mann–Whitney *U* test was used for $n = 9$ (Input) and 3 (YY1-bound). Source data are provided as a Source Data file.

targeting the genomic regions either directly overlapping rs398206 (gRNA 1, 3, and 4) or ~25 bp upstream (gRNA 2) resulted in 61–82% reduction in *MX2* expression levels ($P = 2.05e-4$–$3.19e-4$, one-sample Wilcoxon test; Fig. 5b), while the same gRNAs do not have effect on nearby *MX1* expression (Supplementary Fig. 15b). As rs398206 is located in the intronic region of *MX2*, it is formally possible that some of the effect on *MX2* expression could be due to physical blocking of passage of transcriptional machinery by the dCAS9-KRAB-MeCP2 system. CRISPRi using dCAS9 without the transcriptional repressor

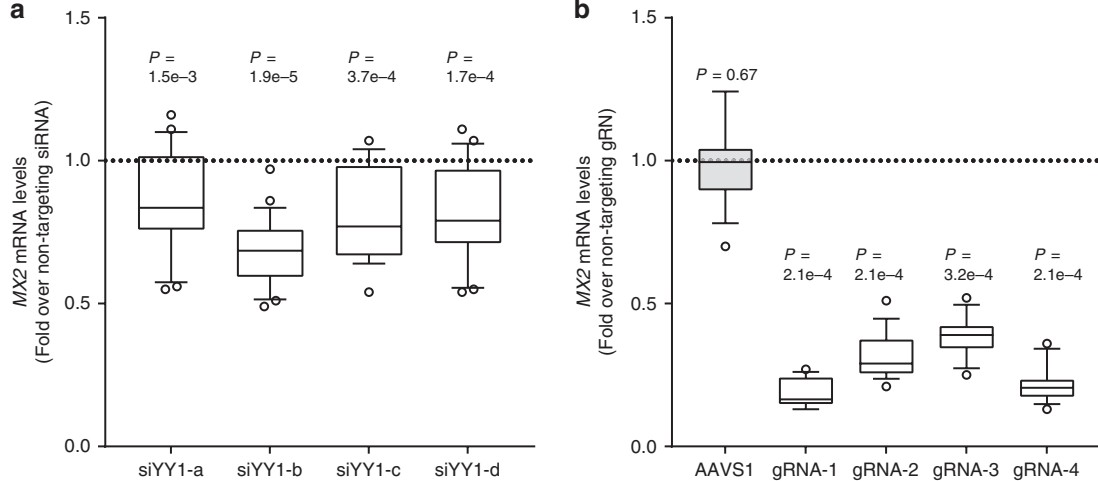

**Fig. 5 YY1 and rs398206 affect _MX2_ expression in melanoma cells. a** YY1 was knocked down using four different siRNAs in UACC647 cells, and _MX2_ levels were measured. _GAPDH_-normalized _MX2_ mRNA levels are shown as fold change over those from non-targeting siRNA. Four biological replicates of $n = 6$ were combined (total $n = 24$). **b** CRISPRi using dCAS9-KRAB-MeCP2 and four different gRNAs targeting rs398206 in UACC903 cells. _MX2_ mRNA levels (_GAPDH_-normalized) are shown as fold change over those from non-targeting gRNA. Three biological replicates of $n = 6$ were combined (total $n = $ 18, except gRNA-3, $n = 17$). gRNA 1, 3, and 4 directly overlap rs398206. gRNA 2 targets ~25 bp upstream of rs398206. AAVS1 (gRNA targeting adeno-associated virus integration site on Chr19). Box: Median and 25th to 75th percentile. Whisker: 10th to 90th percentile. _P_-values are shown from one-sample Wilcoxon test (two-sided) for difference from non-targeting siRNA/gRNA. Dotted line denotes the _MX2_ levels in non-targeting siRNA/gRNA control. Source data are provided as a Source Data file.

elements, however, displayed little or no effect on _MX2_ expression, which is consistent with the CRISPRi effect on _MX2_ being mainly transcriptional (Supplementary Fig. 15c–e).

To identify additional support for rs398206 regulating _MX2_ via YY1, we examined available chromatin interaction data involving YY1. Notably, YY1 was recently shown to mediate chromatin looping and contribute to interactions between gene promoters and enhancers within TADs[39]. Given this, we examined YY1-mediated chromatin interaction around the genomic region encompassing rs398206 in these published Hi-ChIP data using YY1 antibody. In the human colorectal carcinoma cell line, HCT116, the 5Kb bin harboring rs398206 displayed a strong interaction with the adjacent bin encompassing _MX2_ promoter area[39] (PET count = 18, $P = 2.27e{-}80$, hypergeometric test; Supplementary Fig. 16). Together these data determined that rs398206 is a functional variant regulating _MX2_ expression via differential YY1 binding in the Chr21q22.3 melanoma locus.

**MX2 accelerates melanoma formation in zebrafish**. _MX2_ is best known for its function in innate immunity as an HIV-1 restriction gene[40,41]. In GTEx tissue types, the highest _MX2_ expression levels are observed in EBV-transformed lymphocytes, whole blood, and spleen, reflecting its main role in innate immune response as an interferon-stimulated gene (GTEx portal; https://gtexportal.org). On the other hand, a previous study also demonstrated that _MX2_ has cell-autonomous function in the proliferation of HeLa cells without IFNα-mediated induction[42]. In our primary melanocyte dataset, _MX2_ is expressed at a relatively high level (median expression ranked at top 26.5% of all expressed genes) without IFNα stimulation. To assess co-expressed genes and enriched pathways in melanocytes expressing _MX2_ at a higher level, we profiled differentially expressed genes between _MX2_-high (top 25%; $n = 28$) and _MX2_-low (bottom 25%; $n = 28$) melanocytes from 106 individuals. We identified 253 differentially expressed genes in _MX2_-high melanocytes (FDR < 0.01 and |log$_2$ fold difference| >1; Supplementary Data 2), which include many of the top correlated immune-response genes

based on pairwise comparisons of all expressed genes in the full melanocyte set (Supplementary Data 3; Supplementary Table 10). Significantly enriched pathways in these 253 differentially expressed genes included those relevant to cellular immune response as might be expected, but also included those affecting cellular growth and cancer (Fig. 6a; Supplementary Table 11) suggesting a possible melanocyte-specific function of MX2 not limited to immune function. On the other hand, an examination of immune infiltrates in melanomas from TCGA did not provide sufficient evidence for the roles of _MX2_ in tumor immune surveillance other than weak correlations with infiltration of a few cell types (Supplementary Note; Supplementary Fig. 17).

Given the possibility of a melanocyte-specific function of _MX2_, we hypothesized that melanocyte-specific _MX2_ expression might have roles in early events of melanoma formation. To test this hypothesis, we first asked if _MX2_ affects growth of primary melanocytes and melanoma cells in a single culture system. Cell growth assays using the xCELLigence system demonstrated that inducible lentiviral expression of MX2 (2–10-fold induction at 72 h; Supplementary Fig. 18a, b) resulted in slightly decreased growth of both melanoma cells and primary melanocytes at 100 ng/ml of doxycycline treatment, while empty vector trans-duced cells did not show any difference (Fig. 6b, c). To begin to understand what genes and pathways might be affected by increased _MX2_ expression and could potentially underlie the altered melanoma cells/melanocytes growth, we performed RNA-seq analyses on melanocytes over-expressing MX2 (2–10-fold induction at 72 h; Supplementary Fig. 18a, c). Differentially expressed genes in MX2-overexpressing melanocytes compared to controls (158 genes, FDR < 10%; melanocytes from 3 individuals, 3 biological replicates each; Supplementary Data 4) displayed enrichment of pathways relevant to immune response, as well as those involving second messenger mediated kinase signaling and cellular growth, among others (Supplementary Table 12; Fig. 6d). We subsequently examined pathways enriched at a shorter time point after MX2 induction (6 h, melanocytes from one individual, 3 biological replicates) and observed consistent results highlighting intracellular second messenger

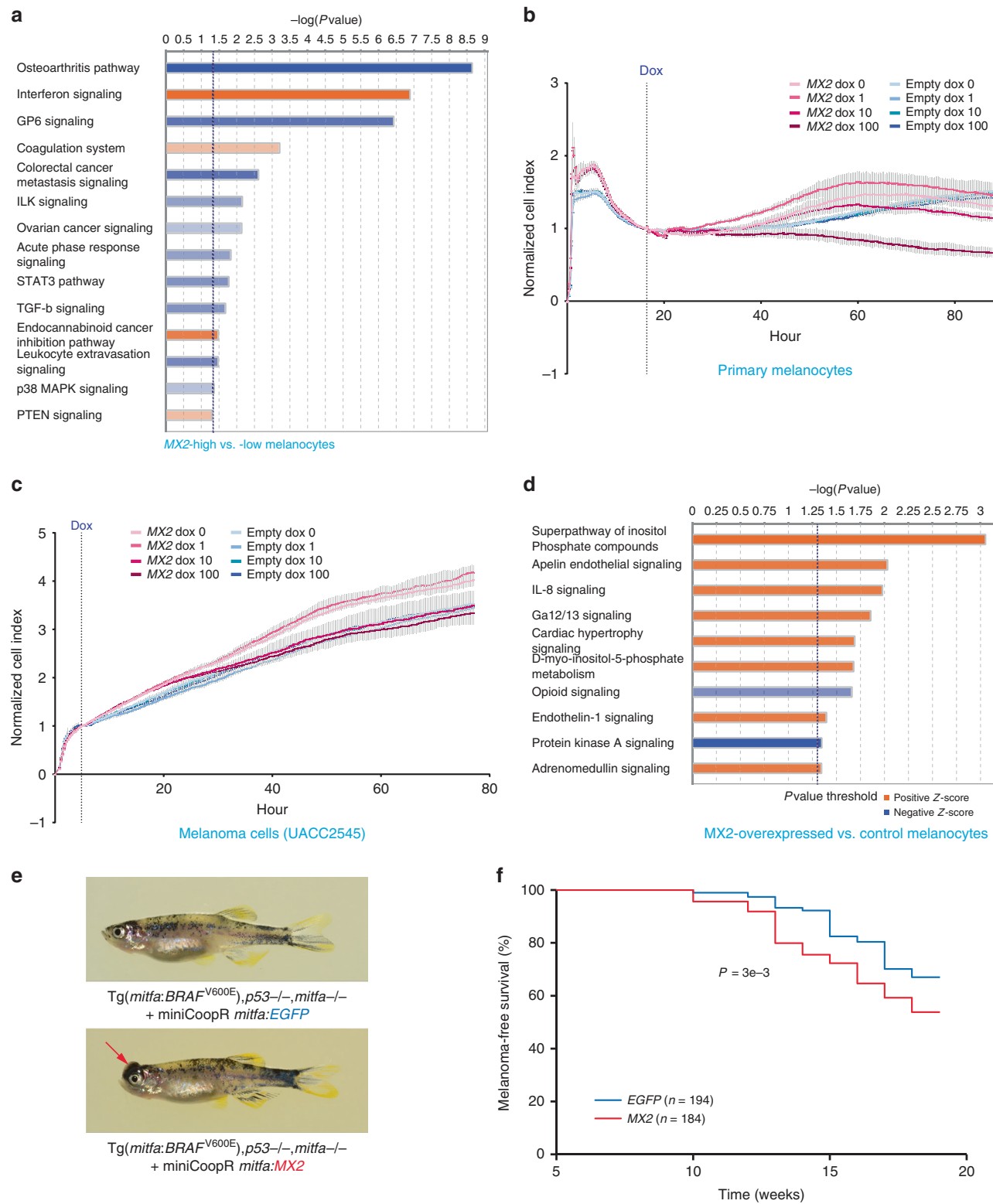

signaling, cancer, cell growth, neurotransmitters, cardiovascular signaling, in addition to cellular immune response (Supplementary Note; Supplementary Data 5 and 6; Supplementary Tables 13 and 14). Since these data did not provide strong support for a specific mechanistic hypothesis linking the effect of increased *MX2* on reduced melanocyte growth in single cultures to its association with melanoma risk, we speculated that the effect of *MX2* on melanocyte growth might change depending on cellular context and microenvironment.

To test this idea and establish a melanocyte-specific role for *MX2* expression in the development or progression of melanoma, we examined transgenic expression of human *MX2* in a zebrafish melanoma model, in conjunction with the most recurrent somatic driver event of melanoma, *BRAF*$^{V600E}$. Using the previously developed miniCoopR transgene system[43], we over-expressed human *MX2* exclusively in the melanocytic-lineage using an *MITF* promoter in the background of *BRAF*$^{V600E}$ and p53$^{-/-}$. The results demonstrated that zebrafish with transgenic human

**Fig. 6 MX2 accelerates melanoma formation. a** Ingenuity Pathway Analysis of differentially expressed genes from *MX2*-high vs. *MX2*-low melanocytes from 106 individuals. 252 differentially expressed genes (FDR < 1% and >2-fold change) between *MX2*-high and *MX2*-low melanocytes (top and bottom quantile based on *MX2* levels; $n = 28$ each) from 106 individuals were used as input for the analysis. Enrichment *P*-values are based on a two-sided Fisher's exact test with no multiple-testing correction applied. (B-C) Cell growth and movement of human primary melanocytes C23 (**b**) or melanoma cell line UACC2545 (**c**) infected with an inducible lentiviral construct of *MX2* cDNA or Empty pINDUCER20 vector were measured on xCELLigence system. Cell Index values were normalized relative to those at the time of doxycycline addition (dotted vertical line: Dox). The amount of doxycycline (dox) is shown in ng/ml and color-coded. Mean Normalized Cell Index (colored dots) and SD (gray vertical lines) are plotted ($n = 3$). A representative set of three biological replicates is shown. **d** Ingenuity Pathway Analysis of differentially expressed genes from RNA sequencing of *MX2* overexpressed vs. control melanocytes from 3 individuals. 158 differentially expressed genes (FDR < 10%) between *MX2*-overexpressing (100 ng/ml doxycycline) vs. control (no doxycycline) melanocytes using 3 biological replicates for 3 individuals were used as input for the analysis. Significantly enriched canonical pathways (P < 0.05 and |Z-score| >1) are color-coded for the direction of effect relative to *MX2*-high melanocytes (**a**) or *MX2*-overexpressing melanocytes (**d**). Enrichment *P*-values are based on a two-sided Fisher's exact test with no multiple-testing correction applied. A weaker to stronger shade of each color represent the relative magnitude of *Z*-scores: Positive *Z*-score between 1 and 2.646 and negative *Z*-score between −1 and −3.464 (**a**) or positive between 1 and 1.134 and negative between −1.342 and −2.236 (**d**), where lightest red is closer to 1 and lightest blue is closer to −1. (**e**) Representative pictures of adult fish from GFP or MX2 group. Pictures were taken at week 10 post-injection. **f** Melanoma-free survival curves of a zebrafish melanoma model[43] (Tg(*mitfa*: *BRAF*$^{V600E}$), *p53*−/−, *mitfa*−/−). The fish were injected at the one cell stage with either miniCoopR *mitfa:MX2* or miniCoopR *mitfa:EGFP* and monitored weekly for melanoma formation. The percentage of melanoma-free fish was combined from three independent experiments and plotted. Log-rank test was used. Source data are provided as a Source Data file.

*MX2* expression presented an accelerated melanoma formation (46% of fish developed melanoma by 19 weeks; $n = 184$) compared to those with *GFP* controls (33% of fish by 19 weeks; $n = 194$) in this genetic background ($P = 0.003$; log-rank test; Fig. 6e, f). While we did not see significant difference in tumor count and pigmentation between the two groups, we observed larger tumor size in the MX2 group compared to GFP control group at median onset ($P = 0.028$; chi-square test), which is consistent with accelerated tumor formation demonstrated through survival analysis (Supplementary Fig. 19). We further performed histopathology analyses of zebrafish melanomas and did not observe gross difference in tumor morphology and invasion between MX2 and GFP groups (Supplementary Fig. 20a–d). Immunohistochemistry for phospho-Histone H3 also did not display a significant difference in cell proliferation between two groups ($P = 0.47$, two-tailed *t*-test, $n = 5$; Supplementary Fig. 20e–g). Together these data are consistent with *MX2* expression contributing to an increased melanoma risk, however consistent with in vitro data, likely through a mechanism other than increased cell proliferation.

## Discussion

In this study, we adopted an integrative approach combining MPRA with cell-type specific epigenomic and eQTL data to efficiently nominate functional variants and susceptibility genes from 20 known melanoma GWAS loci. We demonstrate that MPRA is a high-throughput variant prioritization tool complementing statistical fine-mapping. While Bayesian fine-mapping methods could nominate a small number of credible causal variants for functional testing, these methods are nonetheless limited by their dependence on imputability, imputation quality, population LD reference, causal assumptions, and choice of functional annotation datasets[44,45]. Further, variants that are tightly linked by LD are often still difficult to distinguish based solely on the genetic data and require individual functional testing. To this end, MPRA provides an agnostic approach to quickly screen a large number of variants linked by LD without relying on assumptions about LD structure or number of causals. By applying a conservative cutoff, we identified 39 variants displaying allelic transcriptional activity from 14 melanoma GWAS loci and showed that they are more likely to change transcription factor binding preference and more likely to be causal compared to the rest of the tested variants. Starting from an average of 52 high-LD melanoma-associated variants per locus that overlap

active chromatin regions, MPRA narrowed the candidate variants down to an average of 2.78 per locus. These results highlight the utility of our approach in complementing statistical fine-mapping and breaking up LD structure using a functional assay. To further prioritize variants functioning through *cis*-regulation of local gene expression, we integrated the 39 MPRA significant variants with melanocyte eQTL data which best represents the cell type of origin for melanoma. By choosing the variants displaying the same allelic directions as those from eQTL, we nominated a short list of candidate genes and variants from four loci. In particular, from the locus on chromosome band 21q22.3, 5 of 22 total tested variants are in near perfect LD ($r^2 > 0.99$) with the GWAS lead SNP (rs408825), including the lead SNP from the original study identifying this locus (rs45430)[9] (Supplementary Table 8). While both Bayesian fine-mapping using PAINTOR in this study and eQTL colocalization using eCAVIAR from our previous study[24] assigned higher probability scores for some of these perfect LD SNPs as causal variants for melanoma risk and/or eQTL, MPRA results for this set of 22 SNPs determined rs398206 ($r^2 = 0.94$ with the lead SNP) as the most significant functional variant. Integrating this result with melanocyte eQTL nominated *MX2* as the best candidate susceptibility gene, and we performed thorough functional characterization of this locus including generating an animal model as a proof of principle.

In addition, our integrative approach efficiently identified the most plausible susceptibility genes and functional variants from three other melanoma GWAS loci. For the melanoma locus on chromosome band 22q13.1, increased *MAFF* levels were correlated with risk. MAFF is a small Maf protein regulated by EGF signaling[46] and plays a role in the oxidative stress response[47], which is relevant to melanomagenesis, given the vulnerability of melanocytes to oxidative stress attributable to melanin production[48]. For the locus on chromosome band 1q21.3 and 2q33-q34, lower *CTSS* and *CASP8* levels were correlated with the risk, respectively. CTSS is a member of cathepsin proteases, initially known as lysosomal enzymes[49]. Increased expression of *CTSS* is correlated with poor prognosis in the context of some cancers (breast and colorectal cancer) but also correlated with better outcome in others (lung cancer). CASP8 is mainly known for its function in apoptosis[50], and GWAS also implicated the *CASP8* locus for breast cancer[51] and basal cell carcinoma[52]. Our results provide strong support for these three genes and warrant further in-depth characterization.

Our approach also successfully identified a small number of candidate functional variants (median 1 variant per locus) in nine

melanoma loci not currently supported by cell-type specific melanocyte eQTLs (Supplementary Table 6). These variants may represent plausible functional candidates for loci where we are underpowered to detect melanocyte cis-eQTLs from only 106 cultures. Alternatively, as our eQTL dataset is based on gene expression patterns captured in cultured primary melanocytes, it is possible that these variants may function within specific cellular contexts not captured in this eQTL dataset. Given that our MPRA assay was conducted in melanoma cells, it may be possible that such variants, for example, may only be functioning in the context of oncogenic signaling (e.g., oncogenic BRAF), tumor suppressor loss (e.g., CDKN2A), or immortalizing events (e.g., mutation of the TERT promoter). Future evaluation of these variants in the context of additional genome-scale datasets, including cell-type specific chromatin interaction data, as well as chromatin features of different cellular contexts, may strengthen or clarify evidence for functionality of leads for these loci.

Through molecular interrogation, we demonstrated that a melanoma-associated intronic variant, rs398206, contributes to allelic expression of MX2 via modifying an enhancer element recruiting the transcription factor, YY1. Our multi-QTL analyses of primary melanocytes further supported a transcriptional mechanism, while ruling out an alternative mechanism through splicing. While our data demonstrated the YY1 transcription factor plays a major role in MX2 regulation via the functional SNP, it is also possible that other trans-acting factors, perhaps among other proteins identified through mass-spec with less pronounced allelic difference (e.g., KLF9, ZBTB7A, PAX3), could also play a role in the observed allelic transcriptional regulation.

Our zebrafish model provided further support for MX2 as a melanoma susceptibility gene accelerating melanoma formation when expressed in the cells of melanocytic-lineage. MX2 has been mainly known as an effector of innate-immunity, conferring restriction to HIV-1 infection[40,41], and its roles in melanoma-genesis have not been studied well. Our findings suggest a role of MX2 in promoting melanoma formation when exclusively expressed in cells of melanocytic-lineage, in the presence of BRAF^V600E, a frequent somatic driver mutation. Unlike the relatively high level of MX2 overexpression in the zebrafish model, increased melanoma risk and MX2 levels associated with this common functional variant (rs398206 risk allele frequency = 0.61, EUR) in human populations are rather modest, which is nearly always the case for cancer-associated common variants. It is possible that the effect of increased MX2 levels have roles only in specific contexts during the process of melanoma development. Our single cell-type growth assays for MX2 also show growth effects for MX2 in melanocytic-lineage in the absence of neighboring cell types from the skin, albeit in the opposite direction. A recent study also reported that MX2 over-expression resulted in decreased growth in primary melanocytes and melanoma cells, while the effect could be the opposite in a subset of melanoma cells[53]. Our zebrafish tumor proliferation assay also suggested that accelerated melanoma formation by MX2 might not be due to increased proliferation per se. Given this apparent complexity, we speculate that MX2 function in the growth of cells of melanocytic lineage might depend on cellular context (e.g., somatic driver events) and or interaction with microenvironment (e.g., immune cells). Further studies will be required to tease out the discrepancy between single cell-type assay and zebrafish model, as well as the molecular mechanisms of how MX2 contributes to melanoma promotion. Nevertheless, our findings established MX2 as a gene displaying pleiotropic roles in melanoma susceptibility and immune response, building on to the established roles of telomere biology (TERT, Chr5p15.33)[19] and oncogene-induced senescence (PARP1, Chr1q42.1)[18] in genetic susceptibility to melanoma in the general population.

## Methods

**Melanoma GWAS fine-mapping.** Fine-mapping of the 20 genome-wide significant loci from the meta-analysis reported by Law and colleagues[1] was conducted following a very similar approach to that of Barrett et al.[54]. Using the results from Law et al.[1] a window was defined as 1 Mb on either side of the most significant variant at each locus. The only exception to this was the region that included the ASIP gene (20q11.2-q12), where a 6 Mb region was instead defined, as this region demonstrated a long-range linkage disequilibrium. Melanoma case/control status was regressed on each genotyped and imputed variant in turn across these regions, with the first four principal components as covariates to account for stratification on 12,419 cases and 14,242 controls from the meta-analysis (only the Harvard GWAS samples and the endometriosis controls from the Q-MEGA_610k study were unavailable). Each region was further narrowed down to the interval covering 500 kb on either side of the most extreme SNPs with p-value <10^{-6} in the initial single SNP analysis and any variants with an imputation INFO score <0.5 (for variants with MAF > = 0.03) or INFO score <0.8 (for variants with MAF < 0.03) was removed. A Bayesian-inspired penalized maximum likelihood approach implemented in HyperLasso[25] was applied to these regions. 100 iterations of HyperLasso were then conducted, using all variants in each region and a Normal Exponential Gamma prior distribution for SNP effects with a shape parameter 1.0[55,56] and scale parameter such that type 1 error is 10^{-4}. Both the study (as a categorical variable) and the first four components were included as covariates. Because of the stochastic nature of the order in which variables are tested for inclusion, this produced a number of potential models, including some that can be considered to 'correspond' to one another, because they differ only by substituting genetic variants that are in very strong LD ($r^2 > 0.9$). By dropping equivalent models, a reduced set of models was produced and was then further reduced by dropping any model whose likelihood was inferior to that of the best model by a factor > = 10. For each remaining model, a logistic regression was conducted using the SNPs in the model to generate adjusted odds ratios. For SNPs retained in any of the models, LD blocks were defined (based on both the HyperLasso results and strength of LD) and the most significant SNP (in a multivariable analysis) from each block was selected. rs36115365 in the region near TERT gene (5p15.33) was not identified in the fine-mapping but included for variant selection as it was identified previously based on functional evidence[19]. Subsequent analysis showed that the risk-associated alleles at rs36115365 and at rs2447853 (the most significant SNP in the region at the time) are in negative LD and when adjusted for the latter SNP, rs36115365 has a P-value of 10^{-4} (Supplementary Table 1). Similarly, for the locus on chromosome band 2p22.2, the optimal model was a 2-SNP, but the secondary signal at rs163094 displayed low INFO scores in some studies rendering imputation less optimal and hence was not used for variant selection. Instead, the best SNP identified by 1-SNP model (rs1056837; a missense variant of CYP1B1) was included as an alternative (Supplementary Table 1). Since the effect of the region around MC1R gene (16q24.3) on melanoma risk is mainly explained by several well-established coding variants[12], we did not include this region in our fine-mapping data.

**MPRA variant selection.** Among 20 genome-wide significant loci from the melanoma meta-analyses by Law and colleagues[1], we prioritized 16 loci where potential cis-regulatory mechanism could be applied. We excluded the other 4 loci containing genes that are implicated in melanoma-associated pigmentation phenotypes (SLC45A2, TYR, MC1R, and ASIP loci), as for many of these genes, coding variants were shown to alter the protein functions. To select high-LD proxy variants for 16 melanoma GWAS loci (Law et al., 2015[1]), we used the following criteria:

1. Primary lead SNPs were taken from Law et al.[1] meta-analysis paper and supplemented by those from additional HyperLasso analysis when there are alternative best SNPs available.
2. For 8 loci, HyperLasso analysis nominated independent multiple secondary signals and these lead SNPs were also added.
3. SNPs of $r^2 > 0.4$ with the primary or secondary lead SNPs using 1000 Genomes phase3 EUR or CEU populations were selected as high-LD variants ($n = 2748$).

To prioritize high-LD variants overlapping melanocyte/melanoma open chromatin regions and/or active promoter/enhancer histone marks, we used one or more of the following criteria:

1. Variant is located within a human melanocyte DHS peak from one or more individuals of three available through ENCODE and Epigenome Roadmap database.
2. Variant is located within a human melanocyte H3K27Ac ChIP-Seq peak from one or more individuals and a H3K4Me1 ChIP-Seq peak from one or more individuals of two and three available through Epigenome Roadmap database, respectively.
3. Variant is located within a human melanocyte H3K27Ac ChIP-Seq peak from one or more individuals and a H3K4Me3 ChIP-Seq peak from one or

more individuals of two and three available through Epigenome Roadmap database, respectively.

4. Variant is located within a human melanoma short-term culture FAIRE-Seq peak from one or more individuals of 11 available from Verfaillie et al.[26].

Based on the above criteria, 832 melanoma GWAS variants were selected to be tested by MPRA. We also included 8 additional variants from Chr1q21.3 that were of $r^2 > 0.8$ with the lead SNP but did not overlap with any functional signature listed above and assigned them as negative controls. Of 832 variants, 306, as well as 8 negative controls were repeated in two libraries to ensure cross-library consistency (see MPRA oligo library design). These 306 variants are also $r^2 > 0.6$ with their lead SNPs and supported by both open chromatin and histone mark annotation from melanocyte or melanoma data. A complete list of variants tested are listed in Supplementary Data 1.

**MPRA oligo library design**. Oligo libraries were designed mainly following the guidelines from published works[21,27] with some modifications. Two libraries containing 32,580 (library 1) and 36,660 (library 2) unique sequence of 200-mer oligos (total of 50,400 unique sequences across two libraries with 18,840 repeated in both) were synthesized by Agilent Technologies (Santa Clara, CA). Composition of each library by GWAS locus and repeated variants are listed in Supplementary Data 1 and Supplementary Table 3. For each variant, 145 bases encompassing the variant with either risk or protective allele in both forward and reverse directions were synthesized together with 10 different 10 base random barcode sequences. These two parts of sequences were separated by recognition sequences for restriction enzymes KpnI (GGTACC) and XbaI (TCTAGA), and flanked by binding sequences for PCR primers (200 bases oligo sequences: 5'-ACTGGCCGCTTCACTG-145 bases-GGTACCTCTAGA-10 bases tag-AGATCGGAAGAGCGTCG-3'). For each variant, a scrambled sequence (core 21 bases encompassing the SNP with the reference allele were shuffled) was also tested in forward and reverse directions in the same manner. This is equivalent to a total of 60 unique sequences designed per variant. When there are additional SNPs other than the test SNP that fall in the 145 bp region, major allele in EUR population was used. For indels, 145 bases length was set based on insertion allele and the deletion allele was left shorter than 145 bases. Random 10 base tag sequences were generated once so that each library has up to 36,660 unique sequences (the same 36,660 tag sequences were used for each library). For the 10 base tag sequence and scrambled 21 base core sequence, only homopolymers of <4 bases were used and the enzyme recognition sites for KpnI, XbaI, and SfiI were avoided. A complete list of oligo sequences can be found in Supplementary Data 7.

**MPRA library construction and sequencing**. MPRA library construction and sequencing was performed following published protocols with some modifications[21,27]. For library cloning, ten femtomole each of gel-purified (10% TBE-Urea polyacrylamide gel) oligo libraries was amplified by emulsion PCR using Herculase II fusion polymerase (Agilent) and 2 μM of primers providing SfiI enzyme sites (Supplementary Table 15), following the instructions of the Micellula DNA Emulsion & Purification Kit (EURx/CHIMERx, Milwaukee, WI). Amplified oligos were quantified using KAPA qPCR assay and verified by DNA sequencing on Ion PGM. Amplicon libraries were prepared using 30 ng of oligos from emulsion PCR using Ion Plus Fragment Library Kit (Thermo Fisher, Waltham, MA) and were sequenced on Ion PGM for an average 203 bp and 175 bp read length at 6.7 million and 5.6 million reads per sample for library 1 and library 2, respectively. To verify oligo library design, 21 bp sequences within oligos including variant and +/− 10 bp were used to map to each sequencing read. Linux command fgrep was used and only 100% sequence match was kept. We then counted the total read depth for each variant represented by the matched sequences, and then calculated the proportion of variant sequences that were verified. For both library 1 and library 2, more than 97% of unique sequences representing the variants in the library were detected with at least 10 sequencing reads. In addition, we found similar proportion and read depth for sequences representing both forward and reverse directions in both libraries. If we use the actual tag sequences as a bait, 82% of tags could be verified, with a caveat that some tags were amplified but not detected because of relatively poor sequencing quality in this position of the amplicon. Sequence-verified oligo libraries were first cloned into pMPRA1 vector (Addgene, Watertown, MA) using SfiI site by electroporation into 10 times higher number of bacterial cells than the number of unique sequences in the oligo library. Cloned pMPRA1 was further digested on KpnI and XbaI sites between 145 bp test sequence and 10 bp barcode sequence, where luc2 ORF with or without a minimal promoter was ligated from pMPRAdonor2 and pMPRAdonor1 (Addgene), respectively. The ligation product was transformed by electroporation into 10 times higher number of bacterial cells in the same manner. Cloned final library for transfection was verified on the gel as a single band after KpnI digestion.

Each library was transfected at least four times (two transfections for each promoter type) into HEK293FT or UACC903 melanoma cells aiming >100 times higher number of transfected cells than the library complexity. The number of transfected cells were estimated using transfection efficiency measured by a separate GFP transfection and visualization. A summary of transfections is listed in Supplementary Table 4. Cells were transfected using Lipofectamine 3000 and harvested at 24 h after transfection for RNA isolation. Total RNA was isolated using Qiagen RNeasy kit (Qiagen, Hilden, Germany), and mRNA was subsequently

isolated using PolyA purist MAG kit (Thermo Fisher). cDNA was then synthesized using Superscript III, from which only short sequences encompassing 10 bp unique barcodes were amplified using Q5 high-fidelity polymerase (NEB, Ipswich, MA) and primers introducing Illumina TruSeq adapter sequences (Supplementary Table 15). Tag sequence libraries were also prepared using input DNA in the same way. Each tag sequence library was sequenced on a single lane of HiSeq2500 (125 bp paired end read).

**MPRA data analyses**. Using FASTQ files from input DNA or RNA transcript sequencing, we counted the number of reads (Illumina read 1) completely matching 10 bp barcode sequences (tag counts) and the same downstream sequence context (TCTAGAATTATTACACGGCG) including an XbaI recognition site and the 3' of the luc2 gene. For each transfection (equivalent to one sequencing run), Tag counts Per Million sequencing reads (TPM) values were calculated by dividing each tag count by the total number of sequence-matching tag counts divided by a million. TPM ratio was then taken as RNA TPM over input DNA TPM and $\log_2$ converted.

From each input DNA library at least 92.1% of barcode sequences were detected, and >89.3% were covered at 10 reads or higher. From RNA samples 87.4–93.3% of barcode sequences were detected, and 84.8–90.8% were covered at 10 reads or higher (Supplementary Table 4). Barcodes showing 10 tag counts or lower were excluded from the further analyses. Median tag counts for the barcodes that were included in the analyses were 48,973–49,903 for DNA input and 46,471–49,758 for RNA output. Reproducibility between transfections were assessed by Pearson correlation of $\log_2$-transformed TPM ratio of each barcode between replicates of transfection. We observed correlation coefficient of 0.944 or higher for each library transfected to HEK293FT cells and 0.935 or higher for UACC903 cells (median Pearson $r = 0.984$; Supplementary Fig. 3). Correlation test was also performed between repeated sequences across libraries. We observed correlation coefficient of 0.821 or higher for HEK293FT cells (Supplementary Fig. 4) and 0.815 or higher for UACC903 cells (Supplementary Fig. 5; median Pearson $r = 0.854$). To avoid low input DNA counts driving variations in RNA/DNA TPM ratios, we removed tags with <6 TPM counts from further analyses. The remaining tags account for 77.47% of all the detected tags (Supplementary Fig. 6).

We analyzed the normalized MPRA measurement ($\log_2$ transformed TPM ratio) using a standard linear regression model. We used the Wald test to test the impact of allele on MPRA level, after adjusting the effect of Strand (forward or reverse direction) as a binary covariate, the effect of Transfection as a categorical covariate with 18 levels (accounting for different promoter status and cell types, as well as cross-transfection variations). To account for the potential heteroskedasticity in the measurement error, we used the robust sandwich type variance estimate in the Wald test as recommended by Long and Ervin[57], and used the R package Sandwich to conduct the analysis. To assess overall transcriptional activity of the 145 bp DNA element including the variant, we used variant-specific scrambled sequences as a null. $\log_2$ transformed TPM ratios of scrambled sequences were regressed against those of either reference or alternative allele while using the same covariates (Strand and Transfection). $\log_2$ TPM ratio for each tag in each transfection was considered as an experimental replicate for regression. The same set of analyses was done only using data from UACC903 melanoma cells and further dropping data for repeated variants from one of two libraries (library 1) to allow subsequent enrichment analyses. Variants showing FDR < 0.01 for both allelic difference and departure from null (for either allele) in both UACC903 only and combined set were called as significant MPRA variants. Complete processed MPRA data can be found in Supplementary Data 7.

**Integration of MPRA variants with melanocyte eQTL variants**. Significant eQTL genes were defined as candidate genes significantly identified by eCAVIAR (CLPP > 0.01) or TWAS (P-value < 0.05/number of genes tested) in melanocytes[24]. Among all 832 variants, variants were selected if they display genome-wide significant melanocyte eQTL P-values (nominal P-value < gene-level genome-wide threshold as defined in Zhang et al.[24]) for one of the significant eQTL genes. Six genes (ARNT, CTSS, PARP1, CASP8, MX2, and MAFF) from five loci had significant eQTL SNPs included in MPRA, and the final MPRA-significant variants overlapped eQTL SNPs in four loci, where a subset of SNPs also displayed the same allelic directions between MPRA and melanocyte eQTL for four genes (CTSS, CASP8, MX2, and MAFF; Supplementary Table 6).

**Motif analysis**. Prediction of variant effects on transcription factor binding sites was performed using the motifbreakR package[28] and a comprehensive collection of human transcription factor binding sites models (HOCOMOCO)[58]. We selected the information content algorithm and used a threshold of 0.001 as the maximum P-value for a transcription binding site match in motifbreakR. $\log_2$ fold change between alternative allele score and reference allele score were used to predict the transcription factor motif effect for each variant.

**Statistical fine-mapping using PAINTOR**. PAINTOR 3.0 (http://bogdan.bioinformatics.ucla.edu/software/paintor) was used to estimate the posterior probability of any SNP within a melanoma locus to be causal. We used default parameters in PAINTOR (window size of 100Kb, max causals 2) and filtered out all

the SNPs with *P*-value >0.5 for computational efficiency. The pairwise LD between all SNPs in each window was computed using the 1000 Genomes EUR data. Functional annotations were provided as part of PAINTOR software which was complimented with a melanocyte specific gene set annotation[24]. In order to determine which annotations are relevant to the phenotype being considered, we ran PAINTOR on each annotation independently and then selected 4 annotations specific to primary melanocytes with high sum of log Bayes factors for the final model to compute trait-specific posterior probabilities for causality. These 4 annotations include melanocyte-specific expressed genes from our melanocyte dataset[24], melanocyte enhancers, transcribed regions, and a histone mark (H3T11ph) from ENCODE and Roadmap. Aside from the variants not meeting our analysis parameters, 462 out of 832 MPRA-tested variants were assigned a posterior probability by PAINTOR and were used for enrichment analyses.

**Melanocyte eQTL, sQTL, and meQTL for the *MX2* locus.** Primary melanocyte eQTL data was obtained from our previous study[24], where 106 individuals mainly of European decent were analyzed. For the marginal eQTL analysis of the genes located in the TAD including rs398206, 21 genes were selected based on expression thresholds of >0.5 by RSEM quantification (RNA-Seq by Expectation Maximization[59]) and ≥6 reads in at least 10 samples. Using FastQTL[60], nominal *P*-value was generated between each gene and all the SNPs +/−2 Mb of rs398206 to test the alternative hypothesis that the slope of a linear regression model between the genotypes and expression levels deviates from 0. The same set of covariates as that used for the eQTL analyses was applied (three top genotype PCs and 10 top PEER factors). A Bonferroni-corrected cutoff of $P < 0.0024$ for 21 genes was then applied to select the genes showing marginal eQTL with rs398206. For sQTL, and meQTL, we performed similar QTL analyses as our previous eQTL study using the same genotype data, population structure covariates, and statistical approaches. We replaced normalized gene expression levels with normalized splice junction events (sQTL), and normalized methylation values (meQTL). We also re-calculated the top 15 PEER factors according to these phenotype values. For sQTL analysis, STAR[61] was used to map the RNA-Seq reads onto the genome (hg19) and then LeafCutter[30] was applied to quantify the splice junctions following the procedures described by the authors (http://davidaknowles.github.io/leafcutter/articles/sQTL.html). For meQTL analysis, we performed genome-wide DNA methylation profiling on Illumina Infinium Human Methylation 450 K BeadChip. Methylation levels of all 106 primary melanocyte samples was measured according to the manufacturer's instruction at Cancer Genomics Research Laboratory at NCI. Measurement of raw methylation densities and quality control were conducted using the RnBeads pipeline[62] and the minfi package[63] (http://bioconductor.org/packages/minfi/). In total, we retained 635,022 probes for the downstream meQTL analysis. No batch effects were identified and there were no plating issues. To obtain the final methylation levels (beta value) for meQTL anlaysis, normalization was performed using the preprocessFunnorm algorithm implemented in minfi R package[63].

**MX2 isoform analysis.** Taqman assays targeting unique junctions of *MX2* transcript isoforms were obtained from Thermo Fisher (Waltham, MA, full-length transcript: Hs01550809_m1 and AP323EZ; ENST00000543692: APYMKKU; ENST00000418103: AP2W9U3). Custom assay design was based on Ensembl75 GRCh37 annotation. RNA was isolated from primary cultures of melanocyte from 106 individuals mainly of European decent[24], and cDNA was synthesized using iScript Advanced cDNA Synthesis Kit (Bio-Rad). Taqman assays were performed in triplicates (technical replicates) to be averaged to single data points and normalized to *TBP* levels. *TBP* was selected among 16 conventional human control genes as being one of the least variable genes in melanocyte dataset based on RNA-seq data.

**Cell culture.** Melanoma cell lines (UACC903, UACC647, UACC2331, UACC502, UACC2545, and UACC612) were grown in the medium containing RPMI1640, 10% FBS, 20 mM HEPES, and Amphotericin B/penicillin/streptomycin. All cell lines were tested negative for mycoplasma contamination.

**Luciferase assays.** For each tested SNP, the exact same 145 bp sequence encompassing rs398206 as tested in MPRA was amplified from genomic DNA of HapMap CEU panel samples carrying either risk or protective allele. Primers were designed to carry 15 base 5' overhangs recognizing either side of pGL4.23 vector after KpnI single cut in both forward and reverse direction to facilitate recombination (Supplementary Table 15). Amplified fragments containing 145 bp sequence were then cloned into pGL4.23 vector using In-Fusion HD Cloning kit (Clontech). The resulting constructs with renilla luciferase into melanoma cell lines (UACC903 and UACC502) using Lipofetamine 2000 reagent following the manufacturer's instructions (Thermo Fisher) in 24-well format. Cells were harvested at 24 h after transfection for luciferase activity assays. All the experiments were performed in at least three biological replicates in sets of 6 replicates.

**EMSA and super-shifts.** Forward and reverse strand of 21-mer DNA oligos encompassing rs398206 were synthesized with 5' biotin labeling (Life Technologies; Supplementary Table 15) and were annealed to make double stranded probes.

Nuclear extracts were prepared from actively growing melanoma cells (UACC2331) using NE-PER Nuclear and Cytoplasmic Extraction Reagents (Thermo Scientific). Probes were bound to 2 μg nuclear extracts pre-incubated with 1 μg poly d(I-C) (Roche, Basel, Switzerland) or 100–750 ng YY1 full-length recombinant protein (31332, Active Motif, Carlsbad, CA) in binding buffer containing 10 mM Tris (pH 7.5), 50 mM KCl, 1 mM DTT, and 10 mM MgCl2 at 4 °C for 30 min. For competition assay, unlabeled competitor oligos were added to the reaction mixture 5 min prior to the addition of probes. Completed reactions were run on 5% or 4–20% native acrylamide gel and transferred blots were developed using LightShift Chemiluminescent EMSA kit (Thermo Scientific) and imaged on Chemidoc Touch (Bio-Rad, Hercules, CA). For antibody-supershifts, 0.6–1.2 μg of antibody against YY1 (sc-1703×, Santa Cruz, Dallas, TX) or rabbit normal IgG (sc-2027, Santa Cruz) were bound to nuclear extract prior to poly d(I-C) (Roche) incubation at 4 °C for 1 h.

**Chromatin immunoprecipitation and genotyping.** Melanoma cells (UACC903, UACC647, and UACC612) were fixed with 1% formaldehyde when ~85% confluent, following the instructions of Active Motif ChIP-IT high sensitivity kit. $7.5 \times 10^6$ cells were then homogenized and sheared by sonication using a Bioruptor (Diagenode) at high setting for 15 min, with 30 s on and 30 s off cycles. Sheared chromatin from $2 \times 10^6$ cells were used for each immunoprecipitation with antibodies against YY1 (sc-1703×, Santa Cruz), or normal rabbit IgG (sc-2027; Santa Cruz) following the manufacturer's instructions. Purified pulled-down DNA or input DNA was assayed by SYBR Green qPCR for enrichment of target sites using primers listed in Supplementary Table 15. A positive control primer set for YY1 binding was designed targeting a known YY1 binding site near *RAF1* gene promoter as reported by Weintraub and colleagues[39]. Relative quantity of each sample was driven from standard curve of each primer set and normalized to 1/100 input DNA. For genotyping rs398206, input DNA or genomic DNA from each cell line, and pulled down DNA from UACC647 cell line (heterozygous for rs398206) was used as template DNA for Taqman genotyping assay (Assay ID: C_2265405_20). All experiments were performed in at least three biological replicates in sets of triplicates.

**Mass spectrometry.** Nuclear lysates for mass spectrometry analysis were collected from UACC903 cells grown in RPMI 1640 media (Gibco) supplemented with 10% FBS, 20 mM HEPES (pH 7.9), 100 U/ml penicillin and 100 μg/ml streptomycin (Gibco)[64]. 21 bp oligonucleotide probes encompassing rs398206 were ordered via custom synthesis from Integrated DNA Technologies with 5'-biotinylation of the forward strand (Supplementary Table 15). Forward and reverse DNA oligos were annealed using a 1.5X molar excess of the reverse strand. DNA pulldowns and on-bead digestion were performed on a 96-well filterplate system as described previously[65]. In short, 500 pmol of annealed DNA oligos were immobilized on 10 μl (20 μl slurry) Streptavidin-Sepharose beads (GE Healthcare, Chicago, IL) for each pulldown. Immobilized DNA oligos were incubated with 500 μg of UACC903 nuclear extract and 10 μg of non-specific competitor DNA (5 μg polydAdT, 5 μg polydIdC). After washing away unbound proteins, beads were resuspended in elution buffer (2 M Urea, 100 mM TRIS (pH 8), 10 mM DTT), alkylated with 55 mM iodoacetamide, and on-bead digested with 0.25 μg trypsin. After desalting using Stage tips, peptides were labeled by stable isotope dimethyl labeling, as described previously[65]. Each pulldown was performed in duplicate and label swapping was performed between replicates to eliminate labeling bias. Matching light and heavy peptides were combined and loaded onto a 30 cm column (heated at 40 °C) packed in-house with 1.8 um Reprosil-Pur C18-AQ (Dr. Maisch, GmbH). The peptides were eluted from the column using a gradient from 9 to 32% Buffer B (80% acetonitrile, 0.1% formic acid) in 114 min at a flow rate of 250 nL/min using an Easy-nLC 1000 (Thermo Fisher). Samples were sprayed directly into a Thermo Fisher Orbitrap Fusion Tribrid mass spectrometer. Target values for full MS were set to 3e5 AGC target and a maximum injection time of 50 ms. Full MS were recorded at a resolution of 120,000 at a scan range of 400–1500 *m/z*. The most intense precursors with a charge state between 2 and 7 were selected for MS/MS analysis, with an intensity threshold of 10000 and dynamic exclusion for 60 s. Target values for MS/MS were set at 2e4 AGC target with a maximum injection time of 35 ms. Ion trap scan rate was set to 'rapid' with an isolation width of 1.6 *m/z* and collision energy of 35%. Scans were collected in data-dependent top-speed mode in cycles of 3 s. Thermo RAW files were analyzed with MaxQuant 1.6.0.1 by searching against the UniProt curated human proteome (released June 2017) with standard settings[66]. Protein ratios were normalized by median ratio shifting and used for outlier calling. An outlier cutoff of 1.5 inter-quartile ranges in two out of two biological replicates was used.

**siRNA knockdown of YY1.** siRNA knockdown of YY1 was performed in the UACC647 melanoma cell line using ON-TARGETplus YY1 siRNAs (J-011796-08, J-011796-09, J-011796-10, and J-011796-11; Dharmacon). Non-targeting siRNA and siRNA targeting *GAPDH* were used for negative and positive control, respectively. Six picomole of siRNA was transfected into $5 \times 10^4$ cells using Lipofectamine RNAiMax (Thermo Fisher) following the reverse transfection procedure in 24-well format. Cells were harvested at 72 h after transfection for RNA isolation. The experiments were performed in 4 biological replicates in sets of 6 replicates.

Total RNA was isolated using RNeasy kit (Qiagen) and cDNA was generated using iScript Advanced cDNA Synthesis Kit (Bio-Rad). *MX2* levels were measured using Taqman probe set (Assay ID: Hs01550809_m1) specifically detecting the full-length isoform and normalized to *GAPDH* levels. qPCR triplicates (technical replicates) were averaged to be considered as one data point. Cells were also harvested for protein isolation from each biological replicate to assess knockdown efficiency by Western blot analysis. Total cell lysates were generated with RIPA buffer (Thermo Scientific, Pittsburgh, PA) and subjected to water bath sonication. Samples were resolved by 4–12% Bis-Tris ready gel (Invitrogen, Carlsbad, CA) electrophoresis. The primary antibodies used were rabbit anti-YY1 (sc-1703×, Santa Cruz), and mouse anti-GAPDH (sc-51907, Santa Cruz). Uncropped gel images and molecular weight marker images for western blot analysis are provided for the corresponding figure in Source Data file.

**CRISPRi of rs398206**. CRISPRi was performed in UACC903 melanoma cell line (AA genotype for rs398206) using four different gRNAs targeting the genomic region on or near rs398206 (gRNA sequences are listed in Supplementary Table 15). Guide RNA target sites were identified using the sgRNA Scorer 2.0 algorithm[67]. Non-targeting gRNA and gRNA targeting the adeno-associated virus site 1 (AAVS1) were used as controls. For each sgRNA, forward and reverse oligonucleotides were annealed and cloned into vector carrying the sgRNA scaffold using the BsmBI restriction enzyme (NEB). For CRISPRi, 400 ng of the vectors containing gRNAs, 500 ng of dCas9-KRAB-MeCP2 (Addgene: 110821) or dCAS9 (Addgene: 47316), and 100 ng of pCMV6-entry vector (carrying neomycin resistance marker) were co-transfected into $2 \times 10^5$ cells using Lipofectamine 2000 (Thermo Fisher) following a reverse transfection procedure scaled to 12-well format. Half the amount of DNA, lipofectamine, and cells were used when conducting 24-well format of culture. Cells were treated with 1 mg/ml Geneticin (Gibco) 24 h after transfection. Cells were harvested 48 h after drug selection for RNA and protein isolation. The experiments were performed in at least 3 biological replicates in sets of 5–6 replicates. Total RNA was isolated using RNeasy kit (Qiagen) and cDNA was generated using iScript Advanced cDNA Synthesis Kit (Bio-Rad). *MX2* levels were measured using Taqman probe set (Assay ID: Hs01550809_m1) specifically detecting the full-length isoform and normalized to *GAPDH* levels. qPCR triplicates (technical replicates) were averaged to be considered as one data point. UACC903 cells tested negative for mycoplasma. Cells were concomitantly transfected and harvested for protein isolation from one representative set of dCAS9 vs. dCas9-KRAB-MeCP2 experiments (Supplementary Fig. 15c, d) for western blotting following the same procedure described before. Proteins were separated on NuPAGE 3–8% Tris-Acetate Protein Gels (Thermo Fisher). The primary antibodies used were mouse anti-CAS9 (7A9–3A3, Active Motif), and mouse anti-GAPDH (sc-51907, Santa Cruz). Uncropped gel images and molecular weight marker images for western blot analysis are provided for the corresponding figure in Source Data file.

***MX2* allele-specific expression**. Melanocyte cells were grown in Dermal Cell Basal Medium (ATCC PCS-200-030) supplemented with Melanocyte Growth Kit (ATCC PCS-200-041) and 1% amphotericin B/penicillin/streptomycin (120-096-711, Quality Biological) as described before[24]. Total RNA was isolated using a miRNeasy Mini kit (217004, Qiagen) further treated with CTAB-Urea following a previously described method[68] to remove excess melanin pigmentation. cDNA was synthesized from total RNA using iScript Advanced cDNA Synthesis Kit (Bio-Rad). Genomic DNA and cDNA were then genotyped for rs398206 using custom Taqman genotyping probe set (ANRWEYM) recognizing both genomic DNA and cDNA (ENST00000543692) with a 5 bp 5′ overhang on the left primer for cDNA based on Ensembl archive 75 annotation. From a total of 44 samples heterozygous for rs398206, 27 samples passing QC (Ct values lower than 38 for both alleles in cDNA and genomic DNA) were used to calculate A/C allelic ratio based on dRn values.

**MX2 over-expression and growth assays**. Melanoma cells and melanocyte growth assays were conducted using lentiviral transduction of *MX2* cDNA under the control of tetracycline-inducible promoter using pINDUCER20 vector (Addgene). The *MX2* cDNA clone (RC206437) in the pCMV6-entry backbone was purchased from Origene and full-length *MX2* cDNA sequence was sub-cloned to pENTR-1A vector by introducing stop codons and removing 3′ Myc-DDK tag before being transferred to pINDUCER20 vector (adapter sequence is listed in Supplementary Table 15). BamHI and MluI sites on pCMV6-entry vector and BamHI and XhoI sites on pENTR-1A were used for sub-cloning. Primary human melanocytes were obtained from Invitrogen and/or the Yale SPORE in Skin Cancer Specimen Resource Core and grown under standard culture conditions using Medium M254 (Invitrogen) with Human Melanocyte Growth Supplement-2 (Invitrogen). For lentivirus production, lentiviral vectors were co-transfected into HEK293FT cells with packaging vectors psPAX2, pMD2-G, and pCAG4-RTR2. Virus was collected two days after transfection and concentrated by Vivaspin20. Cells were incubated with virus for 24 h, followed by drug selection (1 mg/ml Geneticin, Gibco), before being subjected to experimental treatments and assays. For xCELLigence assays, optimized number of cells for each cell type were seeded to RTCA E-plate 16 and grown until the Cell Index stabilized. Varying amounts of doxycycline were then added, and the Cell Index was monitored for 72 h. All experiments were performed in 3 biological replicates in sets of triplicates. For each

round, cells were concomitantly infected and harvested for protein isolation at 72 h of doxycycline treatment to assess MX2 levels by western blotting. The primary antibodies used were rabbit anti-MX2 (NBP1-81018, Novus Biologicals), and mouse anti-GAPDH (sc-51907, Santa Cruz). Uncropped gel images and molecular weight marker images for western blot analysis are provided for the corresponding figure in Source Data file.

**Differentially expressed genes in MX2-high melanocytes**. From the RNA-seq data of primary melanocytes ($n = 106$), we profiled differentially expressed genes (DEGs) between *MX2*-high (top 25%; $n = 28$) and *MX2*-low (bottom 25%; $n = 28$) samples. Total counts of mappable reads for each annotated gene (GENCODE v19) was obtained using featureCounts from Rsubread package[69]. The SARTools[70] workflow was used to perform quality control, apply differential analysis and generate reports based on the count data from both *MX2*-high and *MX2*-low groups. edgeR[71] was selected as the statistical methodology to determine differential expression based on the negative binomial distributions. The final DEG list with criteria FDR < 0.01 and |log$_2$ fold difference| >1 was applied to Ingenuity Pathway Analysis (IPA). We also performed pairwise gene expression correlation between *MX2* and the rest of 37,854 genes expressed in these 106 primary melanocytes. Bonferroni-correction was applied to *P*-values of Pearson correlations ($P < 0.05/37,854$ or $1.32e{-}06$; equivalent to FDR < 0.05). Genes with expression levels significantly correlated (FDR < 0.05, $N = 377$) with those of *MX2* were selected for IPA analysis.

**RNA-seq of melanocytes over-expressing MX2**. For RNA-seq analyses of MX2 over-expressing melanocytes, primary cultures of melanocytes from three individuals (C23, C29, and C53) were selected based on their low basal *MX2* expression levels. Cells were grown and infected with the lentiviral system using *MX2* cDNA cloned into pINDUCER20 or empty vector as described above. Following drug selection, cells were treated with 0 or 100 ng/ml doxycycline (total of three conditions for each cell line: 0 or 100 ng/ml doxycycline for pINDUCER20-*MX2* infected cells, and 100 ng/ml doxycycline treatment for empty vector infected cells) for 72 h before being harvested for RNA and protein isolation. For each cell line, three separate infections (biological replicates) were performed and sequenced for transcriptome analysis (total of 27 samples sequenced: 3 conditions, 3 cell lines, and 3 biological replicates). Western blotting was performed for each cell line to estimate the level of MX2 induction. For a short-term induction experiment in C23 melanocytes, varying induction times (3, 6, 24, and 72 h) were tested using 100 ng/ml doxycycline, and 6 h was selected as a time point for RNA-seq analysis based on MX2 induction levels by western blotting. C23 cells treated with 100 ng/ml doxycycline were collected at 6 h and 72 h timepoints with matched controls without treatment in three biological replicates. Total RNA was isolated in the same way as previously described[24]. Sequencing library was constructed following Illumina TruSeq Standard mRNA Library protocol. 150 bp paired-end sequencing was performed on NovaSeq 6000 to achieve at least 50 million reads per sample (range 53.0–82.4 M). FASTQ raw data was received and quality control was performed by the MultiQC RNA-Seq module[72] (https://multiqc.info). Quasi-mapping algorithm Salmon[73] was used to provide fast and bias-aware quantification of transcript expression using GENCODE human transcripts database (release 29). A principal component analysis was performed based on the expression qualification, and based on the results, differentially expressed genes (DEGs) were calculated with DESeq2[74] adjusting for cell line, biological replicate, and library construction batch. The expression threshold FDR < 0.1 was recognized as DEGs after *MX2* over expression by comparing pINDUCER20-*MX2*-infected cells with (100 ng/ml) or without doxycycline treatment. The list of significant DEGs was analyzed using IPA for pathway enrichment analysis. Threshold of $P < 0.05$ and non-zero z-scores were used for identifying significantly enriched pathways. DEG analysis of cells infected with empty vector followed by 100 ng/ml doxycycline treatment vs. those infected with pINDUCER20-*MX2* with no treatment was performed as a control. IPA analysis using DEGs from this control analysis (1838 genes at FDR < 0.01 cutoff) did not overlap in the same direction of change with the main pathways enriched by MX2 overexpression except for Apelin Endothelial Signaling Pathway (Supplementary Table 12 and 16).

**Zebrafish melanoma model**. The *MX2* open reading frame was cloned under the control of the melanocyte-specific mitfa promoter into the miniCoopR expression vector[43]. Tg(mitfa:BRAF$^{V600E}$), p53−/−, mitfa−/− embryos were injected at the one cell stage with either miniCoopR mitfa:MX2 or miniCoopR mitfa:EGFP (as a negative control). Embryos were sorted for melanocyte rescue at 5 days post fertilization (dpf) and raised to adulthood. Tumor formation was monitored weekly between weeks 10 and 19 post-injection. There were no observable differences between the negative control and *MX2* group in melanocyte rescue efficiency or overall pigmentation of 5 dpf larvae. Representative pictures of adult fish were taken at week 10 post-injection. Three independent experiments of different sample sizes were performed by independent injections of DNA constructs replicating similar results. Kaplan-Meier survival curve was plotted using the combined data from these three sets, and *P*-value was calculated using log-rank test. Pigmentation, size, and numbers of melanomas per fish were qualitatively assessed at median onset in the two groups. Data from the three independent experimental sets were combined and P-values were calculated using

chi-square test. For histology analyses, zebrafish were euthanized at 20 weeks of age, and fixed in 4% paraformaldehyde overnight at 4 °C. Paraffin embedding, sectioning, hematoxylin and eosin (H&E) staining were performed according to standard techniques by the Brigham & Women's Hospital Pathology Core. Immunohistochemistry was performed on the Leica Biosystems Bond III automated staining platform. Phospho-Histone H3 (Ser10) Antibody (#9701, Cell Signaling Technologies, Danvers, MA) was run at 1:500 dilution using the Leica Biosystems Refine Detection Kit with citrate antigen retrieval, after bleaching of the melanin pigment. Histologic sections were imaged using an Olympus BX41 microscope with an Olympus DP70 camera. p-H3 foci were counted on 7 distinct tumor areas per fish for a total surface of 1 mm². Zebrafish were handled humanely according to our vertebrate animal protocol that implements the principles of replacement, reduction, and refinement ('three Rs'), has been approved by Boston Children's Hospital Animal Care Committee, and includes detailed experimental procedures for all in vivo experiments described in this paper.

**Statistical analyses**. All cell-based experiments were repeated at least three times with separate cell cultures. When a representative set is shown, replicate experiments displayed similar patterns. For all plots, individual data points are shown with the median or mean, range (maximum and minimum), and 25th and 75th percentiles (where applicable). The statistical method, number of data points, and number and type of replicates are indicated in each figure legend.

**Reporting summary**. Further information on research design is available in the Nature Research Reporting Summary linked to this article.

## Data availability

The sequencing data generated during this study (MPRA sequencing and RNA-seq data) are deposited in Gene Expression Omnibus (https://www.ncbi.nlm.nih.gov/geo/) as a SuperSeries under the accession number GSE129250 [https://www.ncbi.nlm.nih.gov/geo/query/acc.cgi?acc=GSE129250]. A complete list of oligo sequences for MPRA libraries and complete processed MPRA data can be found in Supplementary Data 7, as well as in the Source Data. Melanocyte eQTL data and RNA-seq expression data from 106 individuals are available through the database of Genotypes and Phenotypes (dbGAP, https://www.ncbi.nlm.nih.gov/gap) under accession number phs001500.v1.p1 [https://www.ncbi.nlm.nih.gov/projects/gap/cgi-bin/study.cgi?study_id=phs001500.v1.p1]. The source data underlying Figs. 1b, 2b–d, 3–5, 6b, c, f and Supplementary Figs. 3–6, 7a, b, 8, 9a, b, d, 10b–f, 11–16, 18, 19a, c, d, and 20g are provided as a Source Data file. The YY1 Hi-ChIP data presented in Supplementary Fig. 16 was download from the NCBI GEO database with accession number GSE99519 [https://www.ncbi.nlm.nih.gov/geo/query/acc.cgi?acc=GSE99519]. Data from the 2015 melanoma GWAS meta-analysis performed by Law and colleagues¹ was obtained and is available from the corresponding authors of that manuscript (Matthew Law, Matthew.Law@qimrberghofer.edu.au; or Mark Iles, M.M.Iles@leeds.ac.uk). All other data is available in the Article, Supplementary Information or available from the authors upon request.

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

## Acknowledgements

The results appearing here are in part based on data generated by the TCGA Research Network (http://cancergenome.nih.gov/). Data were also obtained from the GTEx Portal (V6p) on 2 December 2015 or dbGaP accession phs000424.v6.p1 on 17 December 2015. We would like to thank K. Jones, S. Bass, K. Teshome, S. Brodie, and other members at the National Cancer Institute Cancer Genomics Research Laboratory (CGR) for help with sequencing efforts and xCELLigence assays, H. Kong, and L. Mehl from the National Cancer Institute, Laboratory of Translational Genomics for proofreading the manuscript, and A. Jermusyk, O. Onabajo, L. Jessop, and L. Amundadottir from National Cancer Institute, Laboratory of Translational Genomics for helpful discussions, and S. Loftus and W. Pavan from National Human Genome Research Institute for the help with melanocyte eQTL study. This work has been supported by the Intramural Research Program (IRP) of the Division of Cancer Epidemiology and Genetics, National Cancer Institute, US National Institutes of Health. This work was also supported by grants from the National Institutes of Health (R01 CA10384, P01 CA163222, R01 CA083115), Cancer Research UK (C588/A19167), Institut National du Cancer, France (INCa_5982), Programme Hospitalier de Recherche Clinique (AOM-07-195), Fondation pour la Recherche Médicale (FDT20130928343), and the Howard Hughes Medical Institute Investigator Program (L.I.Z.). The Vermeulen lab is part of the Oncode Institute, which is partly funded by the Dutch Cancer Society (KWF). We also thank all the cohorts, funders, and investigators who contributed to the melanoma GWAS, acknowledged by Law and colleagues[1], from which data was used towards fine-mapping. The content of this publication does not necessarily reflect the views or policies of the US Department of Health and Human Services, nor does mention of trade names, commercial products, or organizations imply endorsement by the US government.

## Author contributions

J.C., T.Z., and K.M.B. conceived and planned the study. J.C., T.Z., L.M.C., M.A.K., K.Y., and B.P. designed and analyzed MPRA assays. J.C., A.V., R.C.H., J.Y., and M.X. conducted experiments for MPRA and MX2 molecular characterization. R.C. designed CRISPR assays. T.Z. and K.M.F. performed data analyses. M.M.M., C.G., and M.V. conducted proteomics analyses. M.B., J.T., B.P., C.H., F.D., J.H.B., M.H.L., and M.M.I. performed fine-mapping of melanoma GWAS data. J.A., H.R., and L.I.Z. designed and performed zebrafish experiments. J.C., T.Z., and K.M.B. wrote the manuscript. K.M.B. and S.J.C. helped supervise the project. A.V. and J.A. contributed equally to this work.

## Competing interests

The authors declare no competing interests.
