## [Peer Review File · Nature Communications]

Reviewers' comments:

Reviewer #1 (Remarks to the Author):

In this manuscript, Choi et al use massively parallel reporter assays and eQTL analysis to fine-map functional variants in 16 melanoma loci and their cis-eQTL genes. They map several variants that affect expression of which four show colocalizing melanocyte cis-eQTLs. They then focus their attention on the MX2 variant and convincingly show how this variant affects expression of MX2 through YY1 binding. Finally, they use a zebrafish melanoma model to show that MX2 expression enhanced formation of tumors. The manuscript is well written and the figures clear and easy to follow. The authors use several novel and complex methods for their analysis but provide flow charts and helpful figures as supplements to aid the reader. Their quantitative MS analysis of proteins binding to a small fragment containing the wild type and variant versions of the rs398206 variant was particularly clever.

The authors do not provide a rationale for how increased expression of MX2 leads to melanoma although they show that it affects the expression of various signaling molecules. However, this lack is hardly a defect and the authors have adequately shown the involvement of these variants in mediating gene expression changes.

I have only a couple of minor suggestions.

1. In figure 1 legend the nature of the red gray and green dots is not explained. What is the difference between the different colors?
2. On page 7, line 170, they discuss 22 variants of which 3 were significant in the MPRA analysis. However, the figure only shows 19 variants and only one is indicated as special. The figure legend explains that the 3 missing SNPs are located further upstream but it would be good to mention it in the text since the reader will spend time looking for them.

Reviewer #2 (Remarks to the Author):

Choi et al. present an original study, aiming at the identification of functionally relevant susceptibility genes for 16 melanoma-associated loci. Massively-parallel reporter assays (MPRA) found 39 candidates with differential allelic transcriptional activity among 832 potentially interesting variants in these loci. Melanocyte-specific expression quantitative trait loci (eQTL) finally pointed to four melanocyte cis-eQTL genes (CTSS, CASP8, MX2, and MAFF), of which the intronic allelic variant rs398206 at the MX2 locus was further characterized in depth by performing a plethora of experimental approaches including zebrafish transgene expression. The manuscript is well written although study design and relevance of the extensive data sets for the findings reported can be argued.

Major points

(1) The authors have invested great efforts in an elegant MPRA discovery study designed to identify functional genetic variants within a large number of genome-wide significant association signals. These efforts are described in great detail in the manuscript and have resulted in 39 variants with significant allelic transcriptional activity. While each one of these variants may be of interest to determine functional susceptibility genes, the authors concentrate specifically on four loci displaying melanocyte eQTL and subsequently on a single locus encompassing the MX2 gene. Little is said about the remaining three melanocyte cis-eQTL genes including CTSS, CASP8, and MAFF. Moreover, there is no additional information on the 35 genetic variants not overlapping an eQTL at the respective locus. This makes it most difficult to accept validity of the integrated approach, even more as the melanocyte eQTL data alone appear to be sufficient to prioritize genetic variants for functionality. The manuscript leaves the reader with the question why preselection of genetic variants via MPRA could provide an advantage over eQTL which by itself

appears a good enough predictor of tissue- or cell-specific gene function. For relevance, the authors should specifically elucidate on the validity of their findings within loci for which no local eQTL was identified. Besides and not related to this issue, the reviewer recognizes MPRA as an effective approach to break up LD structures. This could, however, be a different topic.

(2) Another weak point of the manuscript is its discussion section. A large part of it is consumed by summarizing the results of the MX2 findings while an in-depth risk-benefit analysis of the chosen elaborate approach is missing. It is rather difficult to gauge the advantage/disadvantage of the MPRA approach relative to gene prioritization based on eQTL findings.

Minor points

- (1) Line 28: The authors state the identification of 39 functional variants. Please clarify in the abstract the number of independent loci represented by these variants.
- (2) Line 36: Functional variant rs398206 which has been associated with melanoma by GWAS and which is an eQTL to MX2 shows a minor allele frequency of 39% for the C allele in European individuals. This leaves the melanoma risk-increasing A allele of this variant highly common suggesting that the incidence of melanoma should be high if an increased MX2 expression leads to strongly increased melanoma formation. The authors may consider this aspect in their discussion.
- (3) Lines 40 and 382: The authors claim to have identified a pleiotropic function of MX2, as the gene promotes melanoma formation but is also important for HIV-1 infection (see also line 374). The authors do not further elaborate on possible pathways for melanoma formation. They also do not discuss the role of melanoma formation for MX2 outside of the zebrafish model.
- (4) Line 78: The authors state that the melanocyte eQTL dataset outperform the data on bulk skin tissue. The authors should elaborate on the meaning of "outperform" in this context.
- (5) Line 124, Fig 1: The authors have identified 39 variants with significant allelic transcriptional activity in their MPRA study. Fig 1 displays the 16 loci investigated while it is difficult to recognize the two loci for which no variant was found significant following this approach. To make it easier on the reader, these two loci should be clearly marked.
- (6) Line 136; Supplementary Fig 10A: The Pearson r of the 39 significant MPRA variants is higher than the one in the non-significant group ($r=0.24$ vs. $r=-0.023$) although the P values for both tests are not significant ($P = 0.149$; $P = 0.556$). This should be clarified.
- (7) Line 158: The authors compare their MPRA data with the results from an earlier melanocyte eQTL study. They mention that four loci overlap in both studies. It would be interesting to elaborate on those loci which harbor melanocyte eQTL but fail to reveal significant MPRA data.
- (8) Line 165; Supplementary Table 8: Seven variants were found to be located in three loci, namely 2p22.2, 21q22.3, and 22q13.1. However, locus 2p22.2 is not displayed, instead there is a locus on 2q33-q34. This should be clarified.
- (9) Lines 194-197: The authors investigated the GTEx database to determine in which tissue rs398206 is a "genome-wide significant eQTL" and which genes are regulated by this variant. Hereby, the genome-wide significance threshold remains unclear. Also, the P -value cutoff in the eQTL studies should be given.
- (10) Supplementary Fig 13: In this figure the color light blue is used in two slightly modified versions, namely to mark the "C-protective allele" and to display the "cDNA A/C". For better visual distinction, the authors should consider to use more distinct colors.
- (11) Lines 245-250, Fig 3A: This figure shows that several proteins bind significantly to the probes in mass-spectrometry. The authors only include YY1 in their further analysis as this protein obviously showed the most significant effect. Nevertheless, the authors should discuss the other significant findings from their analysis. Specifically, proteins which bind preferably the C allele of rs398206 could be of interest. Furthermore, color coding in Fig 3A is unclear.
- (12) Lines 257-259, Fig 4A: The authors performed a ChIP analysis to demonstrate the YY1 binding near rs398206. Two cell lines are included into the experiments carrying either the rs398206 genotype AA (UACC903) or the rs398206 genotype AC (UACC647). Inclusion of a cell-line with a CC genotype would be of particular interest. Also, the authors should apply the same scaling for both bar plots in Fig 4A.
- (13) Lines 299-306: The authors investigate melanocytes with high MX2 expression in comparison

to low MX2 expression. They identify 253 differentially expressed genes and conclude that these genes suggest a possible non-immune function of MX2. This raises the question whether the expression of MX2 correlates with the expression of any other gene in the full melanocyte dataset. (14) Lines 305 and 317, Fig 6: In the text the authors first mention Fig 6C and thereafter refer to Fig 6A-B. The order should be corrected in the text or in the composite figure.

(15) Lines 320-324: The authors test if melanocytes which overexpress MX2 show differential gene expression compared to controls. It needs to be clarified whether the 158 genes identified overlap with the genes from the experiments mentioned in lines 299-306.

(16) Lines 427-446: Here the authors detail the general MPRA workflow and also give information about replication rates and results. This is again mentioned later in the Methods section. The authors should avoid redundancy.

(17) Lines 534-537: The sentence reads awkward and should be rephrased.

(18) Line 615: The meaning of RSEM is unclear.

(19) Methods: There are several inconsistencies in naming companies and providing specifics such as country and city.

Reviewer #3 (Remarks to the Author):

The authors implemented an elegant pipeline to identify susceptibility genes for melanoma progression.

Revise please

MAJOR

As a "non-expert in the GWAS studies" I would like to suggest that some details could be clarified/written in a more easy/friendly way. For instance more than having in the abstract... rs398206...why not rs398206 intronic variant...immediately the reader finds the relevant detail...in my opinion...

-It wasn't clear why the authors do not discuss or compare the function of rs398206 intronic variant with the previous described rs45430 (Barrett JH et al. Genome-wide association study identifies three new melanoma susceptibility loci. Nat Genet. 2011;43(11):1108-1113. doi:10.1038/ng.959, Gibbs et al, 2015, DOI: 10.1158/1055-9965) intronic snp...

-RNA seq experiments in MX2 overexpressed vs control melanocytes (Figure6D) was performed after 72h of doxycycline treatment...this is too much time of induction- is of no surprise that this timings will not give much information on the downstream events. This experiment should be repeated with a shorter time point of induction 3-to 6h to narrow down the direct genes and possibly provide a mechanistic hypothesis.

-The results that mostly support the role of MX2 in melanoma progression are definitely the zebrafish melanoma model, however this is the data that is least characterized and analyzed. And the in vitro opposite results further render a more thorough characterization of the MX2 melanoma zebrafish model.

Do not agree with the claims in 376...the in vitro data shows that MX2 reduces growth of melanocytes, but overexpressing MX2 in vivo increases the incidence of melanoma...

Therefore, in my understanding, these results allude to a non-cell autonomous role of MX2...or otherwise please clarify better...

Also the IL8 signalling pathway upregulation may indicate a role of innate immune cells/recruitment of neutrophils to the tumors...

The authors should:

- Show representative images of the fish with melanoma to accompany the melanoma-free survival curves.

- Histopathology characterization of the MX2 melanoma: differentiation / infiltration /metastization

/and pigmentation.

- Quantification of number of events per fish along time – are there more foci – initiation events per animal?
- Quantification of the size of the melanoma lesions (in vitro data show reduced growth – is it possible that there is an increase in the incidence but the melanomas are slow growers?)
- Induce the MX2 melanoma model in innate immune deficient background (IRF8, Runx, Panther or Myd88 mutants, not all but 1 /2 if possible), to investigate a bit further the innate immune non-cell autonomous effect

MINOR

-Supplementary Figure 2 could be included in Figure 1...it would be most helpful for non experts ;)

Figure 6

-Panels of the figures do not appear in the same order of the text, becomes confusing. Starts in C- goes to A/B and then to D...

-line 301 should include “and MX2-low (bottom 25%; n = 28) melanocytes” from 106 individuals

We thank the reviewers for their constructive and helpful comments. We address reviewer comments point-by-point, below. Where significant additions or changes have been made to the main text or supplement, we have highlighted such text in yellow.

Reviewers' comments:

Reviewer #1 (Remarks to the Author):

In this manuscript, Choi et al use massively parallel reporter assays and eQTL analysis to fine-map functional variants in 16 melanoma loci and their cis-eQTL genes. They map several variants that affect expression of which four show colocalizing melanocyte cis-eQTLs. They then focus their attention on the MX2 variant and convincingly show how this variant affects expression of MX2 through YY1 binding. Finally, they use a zebrafish melanoma model to show that MX2 expression enhanced formation of tumors. The manuscript is well written and the figures clear and easy to follow. The authors use several novel and complex methods for their analysis but provide flow charts and helpful figures as supplements to aid the reader. Their quantitative MS analysis of proteins binding to a small fragment containing the wild type and variant versions of the rs398206 variant was particularly clever.

The authors do not provide a rationale for how increased expression of MX2 leads to melanoma although they show that it affects the expression of various signaling molecules. However, this lack is hardly a defect and the authors have adequately shown the involvement of these variants in mediating gene expression changes.

I have only a couple of minor suggestions.

1. In figure 1 legend the nature of the red gray and green dots is not explained. What is the difference between the different colors?

We have now added the description in the legend for different colors of the circles.

“Putative function of 39 significant MPRA variants are shown as activator (red circle), repressor (blue circle), or both (purple circle) (expression levels of either allele is higher, lower, or higher and lower than those of scrambled sequence, respectively). Gray variants above the FDR 1% cutoff are those that failed additional criteria (allelic difference in the combined data or significant departure from the scrambled control). No significant variants were identified from the loci on Chr6p22.3 and Chr7p21.1.”

2. On page 7, line 170, they discuss 22 variants of which 3 were significant in the MPRA analysis. However, the figure only shows 19 variants and only one is indicated as special. The figure legend explains that the 3 missing SNPs are located further upstream but it would be good to mention it in the text since the reader will spend time looking for them.

We agree that the presented numbers in the text and Fig 2A could create confusion. We now clarified this in the text as well as in the legend:

In the **Results**:

“In this locus twenty-two variants were originally tested in MPRA (**Supplementary Table 9**), nineteen of which are located in the first intron of the *MX2* gene (**Fig 2A**), and the remaining three upstream. Of these, three variants were significant in MPRA, and rs398206, in particular, (**Fig 2A**, shown in magenta) displayed the lowest P-value of all 832 tested variants.”

Fig 2A legend:

“Only the 19 variants located in the first intron of *MX2* coming from the primary GWAS signal are shown (the other three from a secondary signal are located upstream of the *MX2* genic region)”

Reviewer #2 (Remarks to the Author):

Choi et al. present an original study, aiming at the identification of functionally relevant susceptibility genes for 16 melanoma-associated loci. Massively-parallel reporter assays (MPRA) found 39 candidates with differential allelic transcriptional activity among 832 potentially interesting variants in these loci. Melanocyte-specific expression quantitative trait loci (eQTL) finally pointed to four melanocyte cis-eQTL genes (CTSS, CASP8, MX2, and MAFF), of which the intronic allelic variant rs398206 at the MX2 locus was further characterized in depth by performing a plethora of experimental approaches including zebrafish transgene expression. The manuscript is well written although study design and relevance of the extensive data sets for the findings reported can be argued.

Major points

(1) The authors have invested great efforts in an elegant MPRA discovery study designed to identify functional genetic variants within a large number of genome-wide significant association signals. These efforts are described in great detail in the manuscript and have resulted in 39 variants with significant allelic transcriptional activity. While each one of these variants may be of interest to determine functional susceptibility genes, the authors concentrate specifically on four loci displaying melanocyte eQTL and subsequently on a single locus encompassing the *MX2* gene. Little is said about the remaining three melanocyte cis-eQTL genes including CTSS, CASP8, and MAFF. Moreover, there is no additional information on the 35 genetic variants not overlapping an eQTL at the respective locus. This makes it most difficult to accept validity of the integrated approach, even more as the melanocyte eQTL data alone appear to be sufficient to prioritize genetic variants for functionality.

The manuscript leaves the reader with the question why preselection of genetic variants via MPRA could provide an advantage over eQTL which by itself appears a good enough predictor of tissue- or cell-specific gene function. For relevance, the authors should specifically elucidate on the validity of their findings within loci for which no local eQTL was identified. Besides and not related to this issue, the reviewer recognizes MPRA as an effective approach to break up LD structures. This could, however, be a different topic.

We appreciate the reviewer's comment. We recognize that we could improve the way we present the data to make our major points clearer to readers, and we have addressed these

points in the new version of the manuscript (see below). Nonetheless, we do believe MPRA is a valid and complementary approach and provides a significant advantage over using eQTL alone for prioritizing GWAS variants for functional study for many loci.

We respectfully disagree that melanocyte eQTL data alone is sufficient to prioritize genetic variants for functionality. While for some loci with very few high-LD variants, eQTL P-values or colocalization probability scores alone might be good predictors of variant *cis*-regulatory functionality, in many cases, even the best available eQTL dataset alone has limitations in prioritizing variants for a few reasons. Similar to Bayesian approaches for GWAS fine-mapping, most eQTL-GWAS colocalization methods rely on major assumptions including presence of a set number of causal variants (usually one), identical LD structures and allele frequencies between GWAS and eQTL populations, and finally all possible causal variants being present in both datasets (Giambartolomei *et al.* 2014 *PLOS Genetics*; Hormozdiari *et al.* *AJHG*, 2016). Imputation accuracy is also critical in correct assessment of probability scores for both GWAS and eQTL fine-mapping, where errors in imputation or missing variants in reference panels (for example, Haplotype Reference Consortium does not include small indels) can compromise the prediction (Chundru *et al.* *Genetics* 2019; Spain and Barrett, *Hum Mol Genet* 2015). Finally, even eQTL datasets of hundreds of samples usually cannot provide resolution that discriminates between tens to hundreds of variants tightly linked by LD better than larger GWAS populations do.

MPRA facilitates comprehensive functional assessment of large numbers of candidate variants and does not rely on assumptions made when fine-mapping via eQTL or colocalization. In addition, MPRA is particularly useful in cases where LD structure does not allow for numerous variants to be distinguished via fine-mapping using GWAS or eQTL data. In fact, our results showed that all five loci displaying eQTL and GWAS signal colocalization initially had 17 to 116 variants per locus displaying genome-wide significant eQTL P-values, which we tested in MPRA and successfully narrowed down to 3 to 9 variants per locus. This summary is now included as a part of **Supplementary Table 7** to make this point clearer. We also added a section in **Methods** to explain the integration of MPRA and eQTL in more details.

(Methods Section) “Integration of MPRA variants with melanocyte eQTL variants”

Significant eQTL genes were defined as candidate genes significantly identified by eCAVIAR (CLPP > 0.01) or TWAS (P-value < 0.05/number of genes tested) in melanocytes (Zhang *et al.* *Genome Research* 2018). Among all 832 variants, variants were selected if they display genome-wide significant melanocyte eQTL P-values (nominal P-value < gene-level genome-wide threshold as defined in Zhang *et al.*) for one of the significant eQTL genes. Six genes (*ARNT*, *CTSS*, *PARP1*, *CASP8*, *MX2*, and *MAFF*) from five loci had significant eQTL SNPs included in MPRA, and the final MPRA-significant variants overlapped eQTL SNPs in four loci, where a subset of SNPs also displayed the same allelic directions between MPRA and melanocyte eQTL for four genes (*CTSS*, *CASP8*, *MX2*, and *MAFF*; **Supplementary Table 7**).

To address the point regarding MPRA-significant variants from those loci not supported by melanocyte eQTL, we added further explanation in the **Discussion** sections as well as a modification in **Supplementary Table 7**. While more than half (23 of 39) MPRA-significant variants belong to five loci with colocalizing eQTL genes, the remaining 16 belong to nine loci

without eQTL support. There are multiple potential explanations for the lack of melanocyte eQTL support in these nine loci. Firstly, compared to many GTEx tissues, for example, our melanocyte eQTL dataset is of modest sample size ($n = 106$), and eQTL variants of smaller effect size and/or low allelic frequencies might not have been detected. Secondly, genetic control of gene expression could be context-dependent, and variants that may be functional only in the context of melanocyte transformation, for example, might not be detected as significant eQTLs in proliferating normal human primary melanocytes. Of note, due to the need for high transfection efficiency, our MPRA assay used melanoma cell lines as opposed to primary melanocytes (which are considerably more difficult to transfect). Thus, MPRA may be detecting context-dependent variants that do not strongly drive allelic activity in primary melanocytes in the absence of *trans*-acting factors.

While we acknowledge that MPRA findings from loci with colocating eQTL genes are clearly stronger leads, we nonetheless contend that these 16 variants from loci without eQTLs make strong candidates for further study, as we successfully narrowed down functional candidates from 3 to 129 high-LD variants to 1 to 5 MPRA-significant variants in each locus. This in fact highlights the utility of MPRA as a prioritization tool complementing statistical fine-mapping (e.g. PAINTOR) in the absence of eQTL support. Changes to the manuscript in **Discussion**:

“Our approach also successfully identified a small number of candidate functional variants (median 1 variant per locus) in nine melanoma loci not currently supported by cell-type specific melanocyte eQTLs (**Supplementary Table 7**). These variants may represent plausible functional candidates for loci where we are underpowered to detect melanocyte *cis*-eQTLs from only 106 cultures. Alternatively, as our eQTL dataset is based on gene expression patterns captured in cultured primary melanocytes, it is possible that these variants may function within specific cellular contexts not captured in this eQTL dataset. Given that our MPRA assay was conducted in melanoma cells, it may be possible that such variants, for example, may only be functioning in the context of oncogenic signaling (e.g. oncogenic BRAF), tumor suppressor loss (e.g. *CDKN2A*), or immortalizing events (e.g. mutation of the *TERT* promoter). Future evaluation of these variants in the context of additional genome-scale datasets, including cell-type specific chromatin interaction data as well as chromatin features of different cellular contexts, may strengthen or clarify evidence for functionality for leads for these loci.”

MPRA is the first step to functional characterization of potential causal variants and genes from GWAS loci. Fundamental validation for trait-associated variants and genes involves complex functional assays including context-specific cell-based assays and animal model analyses, which are beyond the scope of the current manuscript. In this manuscript we provide a reasonable prioritization scheme and present a proof of principle example for the most plausible locus (*MX2*). While we cannot provide the same level of functional validation for *CTSS*, *CASP8*, and *MAFF* loci in this manuscript, our results in *MX2* suggest that the other three are likely strong candidates for in-depth functional studies in the future.

(2) Another weak point of the manuscript is its discussion section. A large part of it is consumed by summarizing the results of the *MX2* findings while an in-depth risk-benefit analysis of the chosen elaborate approach is missing. It is rather difficult to gauge the advantage/disadvantage

of the MPRA approach relative to gene prioritization based on eQTL findings.

Based on the reviewer's input, we added a description in the **Discussion** section to further elaborate the utility of MPRA approach while also addressing the major point (1) shown above.

"We demonstrate that MPRA is a high-throughput variant prioritization tool complementing statistical fine-mapping. While Bayesian fine-mapping methods could nominate a small number of credible causal variants for functional testing, these methods are nonetheless limited by their dependence on imputability, imputation quality, population LD reference, causal assumptions, and choice of functional annotation datasets^{45,46}. Further, variants that are tightly linked by LD are often still difficult to distinguish based solely on the genetic data and require individual functional testing. To this end, MPRA provides an agnostic approach to quickly screen a large number of variants linked by LD without relying on assumptions about LD structure or number of causals. By applying a conservative cutoff, we identified 39 variants displaying allelic transcriptional activity from 14 melanoma GWAS loci, and showed that they are more likely to change transcription factor binding preference and more likely to be causal compared to the rest of the tested variants. Starting from an average of 52 high-LD melanoma-associated variants per locus that overlap active chromatin regions, MPRA narrowed the candidate variants down to an average of 2.78 per locus. These results highlight the utility of our approach in complementing statistical fine-mapping and breaking up LD structure using a functional assay. To further prioritize variants functioning through *cis*-regulation of local gene expression, we integrated the 39 MPRA significant variants with melanocyte eQTL data which best represents the cell type of origin for melanoma. By choosing the variants displaying the same allelic directions as those from eQTL, we nominated a short list of candidate genes and variants from four loci. In particular, from the locus on chromosome band 21q22.3, 5 of 22 total tested variants are in near perfect LD ($r^2 > 0.99$) with the GWAS lead SNP (rs408825), including the lead SNP from the original study identifying this locus (rs45430)⁹. While both Bayesian fine-mapping using PAINTOR in this study (**Supplementary Table 9**) and eQTL colocalization using eCAVIAR from our previous study²⁴ assigned higher probability scores for some of these perfect LD SNPs as causal variants for melanoma risk and/or eQTL, MPRA results for this set of 22 SNPs determined rs398206 ($r^2 = 0.94$ with the lead SNP) as the most significant functional variant. Integrating this result with melanocyte eQTL nominated *MX2* as the best candidate susceptibility gene, and we performed thorough functional characterization of this locus including generating an animal model as a proof of principle."

Minor points

(1) Line 28: The authors state the identification of 39 functional variants. Please clarify in the abstract the number of independent loci represented by these variants.

We have now added the number of independent loci in the abstract:

"Starting from 16 melanoma loci, we selected 832 variants overlapping active regions of chromatin in cells of melanocytic lineage and identified 39 candidate functional variants **from 14 loci** displaying allelic transcriptional activity by MPRA"

(2) Line 36: Functional variant rs398206 which has been associated with melanoma by GWAS and which is an eQTL to MX2 shows a minor allele frequency of 39% for the C allele in European individuals. This leaves the melanoma risk-increasing A allele of this variant highly common suggesting that the incidence of melanoma should be high if an increased MX2 expression leads to strongly increased melanoma formation. The authors may consider this aspect in their discussion.

Given the identified functional variant has a modest OR (OR = 1.15; Law *et al. Nature Genetics* 2015) for melanoma risk and high population allele frequency (MAF=0.39, EUR) as is typical for a GWAS finding, we are not implying that this single variant alone can accelerate melanoma formation in human populations. Similarly, the MX2 eQTL for this SNP in melanocytes is significant, but the allelic fold difference, like in most eQTLs, is not large (slope = 0.7; P = 6.6e-15), while MiniCoopR system in zebrafish expresses human genes at relatively high levels. Given this, one would perhaps expect more subtle effects on melanoma development in humans than observed in the zebrafish model. It is also possible that this variant could contribute to aspects of melanomagenesis in a specific context where increased MX2 could function in concert with other events (e.g. *BRAF*^{V600E}, UV exposure, etc.) to accelerate tumor development. We have now added this point in the **Discussion**:

“Unlike the relatively high level of MX2 overexpression in the zebrafish model, increased melanoma risk and MX2 levels associated with this common functional variant (rs398206 risk allele frequency = 0.61, EUR) in human populations are rather modest, which is often the case for cancer-associated common variants. It is possible that the effect of increased MX2 levels have roles only in specific contexts during the process of melanoma development.”

(3) Lines 40 and 382: The authors claim to have identified a pleiotropic function of MX2, as the gene promotes melanoma formation but is also important for HIV-1 infection (see also line 374). The authors do not further elaborate on possible pathways for melanoma formation. They also do not discuss the role of melanoma formation for MX2 outside of the zebrafish model.

While we could not conclusively remark on the possible pathways based on the data we have from the zebrafish model and the cell-based assays, we elaborated a bit more on the complexity of MX2 roles in the **Discussion**. Complexity of MX2 roles in melanocyte/melanoma growth was also consistently observed by a recent publication (Juralviciute *et al. Pigment Cell Melanoma Res* 2019) that was made available during the revision of our manuscript, where MX2 had opposite effects on increased/decreased cell growth depending on melanoma cell lines that were tested. Further studies will be required to tease out the molecular mechanisms and involved biological pathways (melanocyte biology-specific versus role in immune function, for example) for MX2 functions in the context of melanoma promotion. Changes to the **Discussion**:

“Our single cell-type growth assays for MX2 also show growth effects for MX2 in melanocytic-lineage without the presence of neighboring cell types, albeit in the opposite direction. A recent study also reported that MX2 over-expression resulted in decreased growth in primary melanocytes and melanoma cells, while the effect could be the opposite in a subset of melanoma cells (Juralviciute et al PCMR 2019). Given the apparent complexity of MX2 function in cell growth of melanocytic lineage, further

studies will be required to tease out the molecular mechanisms of how MX2 contributes to melanoma promotion.”

(4) Line 78: The authors state that the melanocyte eQTL dataset outperform the data on bulk skin tissue. The authors should elaborate on the meaning of “outperform” in this context.

Through colocalization using eCAVIAR and TWAS methods, melanocyte eQTL (n = 106) alone identified candidate susceptibility genes for six melanoma-associated loci and further identified four novel loci (Zhang *et al. Genome Research* 2018). Importantly, all four of the novel loci were later identified as genome-wide significant melanoma GWAS loci in the new and larger GWAS meta-analysis or as a significant melanoma and nevus count pleiotropic locus (Landi *et al., Nature Genetics, in press*; Duffy *et al., Nature Communications*, 2018). On the other hand, two types of bulk skin eQTL datasets from GTEx (n = 302 and 196) together identified genes for 5 loci and no novel loci. These results highlight that melanocyte eQTL is efficient in identifying candidate susceptibility genes from melanoma GWAS loci (twice more loci compared to bulk skin tissue-based eQTLs of larger sample sizes) that are also likely to represent *cis*-regulation in melanocytic lineage as opposed to those that are common in many cell types in bulk skin tissue.

We edited the indicated sentence to clarify the meaning in **Introduction** (line 81):

“This dataset identified more candidate susceptibility genes than using eQTLs from datasets of larger sample size generated from bulk skin tissues, other tissue types from GTEx, and melanoma tumors, highlighting the utility of cell-type specific eQTL dataset for functional follow-up of GWAS regions”

(5) Line 124, Fig 1: The authors have identified 39 variants with significant allelic transcriptional activity in their MPRA study. Fig 1 displays the 16 loci investigated while it is difficult to recognize the two loci for which no variant was found significant following this approach. To make it easier on the reader, these two loci should be clearly marked.

Following the reviewer’s suggestion, we clearly marked the two loci (6p22.3 and 7p21.1) with no variants meeting our criteria in the figure legend. All 39 significant variants from 14 other loci are also clearly marked by color coding (non-gray). Note that we are displaying in this figure the P-values and effect sizes from melanoma cell line data (indicated in the figure legend), while the 39 variants were identified using both 1) allelic difference in combined dataset of two cell lines and 2) significant departure from the scrambled control level for both alleles (activator or repressor).

(6) Line 136; Supplementary Fig 10A: The Pearson r of the 39 significant MPRA variants is higher than the one in the non-significant group (r=0.24 vs. r=-0.023) although the P values for both tests are not significant (P = 0.149; P = 0.556). This should be clarified.

Changes in bold:

“We then asked if the observed allelic differences from MPRA are in part due to differential binding of transcription factors. For this, we predicted allelic transcription factor binding affinity of each tested variant using motifbreakR **and subsequently correlated the predicted allelic binding scores with the allelic transcriptional**

activities measured from MPRA. Notably, the MPRA-significant variants displayed a higher level of correlation compared to that of non-significant ones, shown by a larger Pearson r (Pearson r = 0.24 vs. -0.023; Supplementary Fig 9A), while the P-values for the both tests are not significant (P = 0.149 and 0.556, respectively)."

(7) Line 158: The authors compare their MPRA data with the results from an earlier melanocyte eQTL study. They mention that four loci overlap in both studies. It would be interesting to elaborate on those loci which harbor melanocyte eQTL but fail to reveal significant MPRA data.

We added a description of the loci with melanocyte eQTL but no MPRA results in the **Results** section. Of six loci identified through eQTL, only 5 were tested in MPRA excluding one locus on Chr5p13.2 where a coding variant of pigmentation gene, *SLC45A2*, mostly accounts for the risk. For the other locus, we added the following explanation:

"For 1q42.12, all the eQTL significant variants did not pass MPRA significance cutoff, and therefore were not prioritized."

(8) Line 165; Supplementary Table 8: Seven variants were found to be located in three loci, namely 2p22.2, 21q22.3, and 22q13.1. However, locus 2p22.2 is not displayed, instead there is a locus on 2q33-q34. This should be clarified.

We apologize for the typo in referencing the relevant locus. It should be 2q33-q34 and was corrected in the text accordingly. We also corrected the same mistake in another sentence from the same paragraph:

"For the purpose of nominating the most plausible candidates for functional follow-up, we mainly focused on five of these six loci with melanocyte eQTL support (1q21.3, 1q42.12, 2q33-q34, 21q22.3, and 22q13.1) that were also tested in our MPRA."

"Similarly, two to three variants each (rs2349075, rs529458487, rs398206, rs408825, rs4383, rs4384, and rs6001033) from three other loci (2q33-q34, 21q22.3, and 22q13.1) also overlapped with melanocyte eQTLs, where lower *CASP8*, higher *MX2*, and higher *MAFF* levels were correlated with melanoma risk, respectively (**Supplementary Table 8**)."

(9) Lines 194-197: The authors investigated the GTEx database to determine in which tissue rs398206 is a "genome-wide significant eQTL" and which genes are regulated by this variant. Hereby, the genome-wide significance threshold remains unclear. Also, the P-value cutoff in the eQTL studies should be given.

We searched rs398206 in the GTEx portal based on GTEx Release V6p, which listed all significant variant-gene pairs associated with eGenes. eGenes were defined using an FDR \leq 0.05 cutoff. To identify all significant variants, a nominal P-value threshold was set for each gene based on a genome-wide empirical P-value as described in the GTEx Release V6p document. We added a short description for this in the text as shown below:

“..no other gene displayed a genome-wide significant eQTL with rs398206 (using a nominal P-value threshold set for each gene based on a genome-wide empirical P-value as defined by GTEx V6p; GTEx portal; <https://gtexportal.org>)”

(10) Supplementary Fig 13: In this figure the color light blue is used in two slightly modified versions, namely to mark the “C-protective allele” and to display the “cDNA A/C”. For better visual distinction, the authors should consider to use more distinct colors.

As suggested, we changed the color of “DNA A/C” to avoid confusion in the new **Supplementary Fig 12**.

(11) Lines 245-250, Fig 3A: This figure shows that several proteins bind significantly to the probes in mass-spectrometry. The authors only include YY1 in their further analysis as this protein obviously showed the most significant effect. Nevertheless, the authors should discuss the other significant findings from their analysis. Specifically, proteins which bind preferably the C allele of rs398206 could be of interest. Furthermore, color coding in Fig 3A is unclear.

To incorporate the reviewer’s comment, we added the following in **Discussion** section:

“While our data demonstrated the YY1 plays a major role in *MX2* regulation via the functional SNP, it is also possible that other *trans*-acting factors, perhaps among other proteins identified through mass-spec with less pronounced allelic difference (e.g. KLF9, ZBTB7A, PAX3), could also play a role in the observed allelic transcriptional regulation.”

We have also clarified the color-coding in the **Figure 3 legend**:

“Proteins passing the inter-quartile > 3 cutoff for both axes are color-coded for consistent (red circle) and inconsistent (yellow circle) direction from label-swapping. Names of consistently identified proteins are shown.”

(12) Lines 257-259, Fig 4A: The authors performed a ChIP analysis to demonstrate the YY1 binding near rs398206. Two cell lines are included into the experiments carrying either the rs398206 genotype AA (UACC903) or the rs398206 genotype AC (UACC647). Inclusion of a cell-line with a CC genotype would be of particular interest. Also, the authors should apply the same scaling for both bar plots in Fig 4A.

Following the reviewer’s suggestion, we included results from a cell line carrying CC genotype (UACC612; a representative result of three biological replicates). We now added these results to **Fig 4A** applying the same scaling for all three bar plots. Unlike AA or AC cell line, we did not observe enrichment of YY1 binding on top of rs398206 that is distinct from neighboring regions.

As supporting evidence that a weaker binding enrichment in this CC line is not due to a defective YY1 ChIP process, we performed internal validation of YY1 ChIP process in UACC612 cell line by showing positive YY1 binding in a known locus (Weintraub *et al*, *Cell* 2017) from the same ChIP DNA (**Supplementary Fig14A**). Direct confirmation of genotypes for AA and CC cell line DNAs used in the ChIP-qPCR was also performed (**Supplementary Fig 14B-C**).

While it should be noted that YY1 binding in these three cell lines could be affected by potential cell line-specific factors other than rs398206 genotype (e.g. DNA copy number changes, accessibility of the chromatin in the region encompassing rs398206, YY1 availability, and variability in formaldehyde fixing and chromatin shearing efficiency), our new data including all three genotypes provide additional support to rs398206 allele-specific YY1 binding in melanoma cells. We note that a better controlled assessment of allele-specific binding is the ChIP-genotyping we previously performed in a heterozygous cell line (**Fig4B-C**) where cell line-specific factors are the same for both alleles. We therefore still emphasize these genotyping data as providing stronger support for allele-specific binding compared to the ChIP results from three cell lines. **Result** text was edited accordingly.

“We subsequently performed chromatin immunoprecipitation (ChIP) using anti-YY1 antibody and scanned ~5Mb genomic region encompassing rs398206 in three melanoma cell lines representing each genotype of rs398206 (AA, AC, and CC). We observed prominent enrichment of YY1 binding on top of rs398206 in AA cell line, a weaker but clear enrichment in AC line, and even weaker binding enrichment over rs398206 in the CC line (**Fig 4A; Supplementary Fig 14**). Given that differences between cell lines (e.g. DNA copy number differences, accessibility of chromatin in the region encompassing rs398206, YY1 levels, and variability in formaldehyde fixing and chromatin shearing efficiency) may also contribute to differential YY1 binding, we also assessed allele-specific YY1 binding in the heterozygous AC cell line (UACC647).”

(13) Lines 299-306: The authors investigate melanocytes with high MX2 expression in comparison to low MX2 expression. They identify 253 differentially expressed genes and conclude that these genes suggest a possible non-immune function of MX2. This raises the question whether the expression of MX2 correlates with the expression of any other gene in the full melanocyte dataset.

Following the reviewer’s suggestion, we performed pairwise correlation analyses between *MX2* and 37,854 genes that are expressed in primary melanocytes (n = 106). At FDR < 0.05 cutoff, 377 genes displayed positive correlation with *MX2* levels (Pearson coefficient > 0.44) and no gene displayed negative correlation (**Supplementary Table 12**). Canonical pathways enriched in these 377 correlated genes clearly indicated cellular immune response and cytokine signaling pathways (**Supplementary Table 13**) as expected from the known function of *MX2* in innate immune response. Half of the top 50 correlated genes (and all of the top 10 correlated genes) are also included in 253 differentially expressed genes (**Supplementary Table 11**), suggesting that those 253 genes capture key immune-response genes highly correlated with *MX2* levels.

In the section that the reviewer pointed out, we do observe that several immune response-related pathways are enriched in 253 differentially expressed genes. However, we also see enriched pathways for cellular growth, cancer and others. Our conclusion from this analysis as well as that of *MX2*-overexpressing melanocytes was that perhaps *MX2* function in melanoma promotion may not be solely attributable to its known function in innate immunity but might also involve diverse cellular processes of melanocytes. Our zebrafish model and cell-based growth assays are not designed to formally test melanocyte-specific expression of *MX2* on immune cells, or alternatively differential expression in immune cells *per se* as we exclusively overexpress *MX2* in melanocytes. While we need further study to tease out immune vs non-

immune functions, all we could conclude from our melanocyte expression data is that we cannot rule out non-immune melanocyte-specific roles of MX2. **We edited the text to make this point more pronounced.**

“To assess co-expressed genes and enriched pathways in melanocytes expressing MX2 at a higher level, we profiled differentially expressed genes between MX2-high (top 25%; n = 28) and MX2-low (bottom 25%; n = 28) melanocytes from 106 individuals. We identified 253 differentially expressed genes in MX2-high melanocytes (FDR < 0.01 and |log₂ fold difference| > 1; Supplementary Table 11), which include many of the top correlated immune-response genes based on pairwise comparisons of all expressed genes in the full melanocyte set (**Supplementary Table 12-13**). Significantly enriched pathways in these 253 differentially expressed genes included those relevant to cellular immune response as might be expected, but also included those affecting cellular growth and cancer (**Fig 6A; Supplementary Table 14**) suggesting a possible melanocyte-specific function of MX2 not limited to immune function.”

(14) Lines 305 and 1317, Fig 6: In the text the authors first mention Fig 6C and thereafter refer to Fig 6A-B. The order should be corrected in the text or in the composite figure.

The panels are re-ordered now.

(15) Lines 320-324: The authors test if melanocytes which overexpress MX2 show differential gene expression compared to controls. It needs to be clarified whether the 158 genes identified overlap with the genes from the experiments mentioned in lines 299-306.

Following the reviewer’s suggestion, we now include a column in the **Supplementary Table 15** to show if any genes differentially expressed in MX2-overexpressed melanocytes are also observed as differentially expressed genes in MX2-high melanocytes. At the gene level, we only see an overlap of three genes including MX2, where MX2 and DUSP8 show the same direction of change but FABP3 shows the opposite direction of change. As our MX2 overexpression experiment is designed to achieve a moderate induction of MX2 (~2.7-fold induction) to best represent the subtle effect of MX2 eQTL including all 106 individuals, we see much lower number of genes being affected by MX2 induction with relatively small effect sizes. On the other hand, 253 differentially expressed genes in MX2-high melanocytes shown in **Supplementary Table 11** are based on the subset of melanocyte lines displaying extreme levels of MX2 (n = 28 in each group) and hence exhibiting much larger effect sizes in general (e.g. 28-fold difference in MX2 levels between high and low groups). Despite the scarcity of overlap in the gene level, both sets of experiments as well as the additional RNAseq data from a shorter MX2 induction (6hr; **Supplementary Table 19 and 20**) share a subset of bigger pathway categories including cellular immune response and cancer.

(16) Lines 427-446: Here the authors detail the general MPRA workflow and also give information about replication rates and results. This is again mentioned later in the Methods section. The authors should avoid redundancy.

We removed the redundant part of the methods (MPRA workflow section) from the text.

(17) Lines 534-537: The sentence reads awkward and should be rephrased.

The indicated sentence was rephrased as follows:

“Each library was transfected at least four times (two transfections for each promoter type) into HEK293FT or UACC903 melanoma cells aiming > 100 times higher number of transfected cells than the library complexity. The number of transfected cells were estimated using transfection efficiency measured by a separate GFP transfection and visualization.”

(18) Line 615: The meaning of RSEM is unclear.

We added more information to define RSEM as follows:

“For the marginal eQTL analysis of the genes located in the TAD including rs398206, 21 genes were selected based on expression thresholds of >0.5 by RSEM quantification (RNA-Seq by Expectation Maximization; Bo Li et al. BMC Bioinformatics 2011) and ≥ 6 reads in at least 10 samples.”

(19) Methods: There are several inconsistencies in naming companies and providing specifics such as country and city.

We addressed this in the **Methods** section.

Reviewer #3 (Remarks to the Author):

The authors implemented an elegant pipeline to identify susceptibility genes for melanoma progression.

Revise please
MAJOR

As a “non-expert in the GWAS studies” I would like to suggest that some details could be clarified/written in a more easy/friendly way. For instance more than having in the abstract... rs398206...why not rs398206 intronic variant...immediately the reader finds the relevant detail...in my opinion...

We now specified that rs398206 is an intronic variant in the abstract.

-It wasn't clear why the authors do not discuss or compare the function of rs398206 intronic variant with the previous described rs45430 (Barrett JH et al. Genome-wide association study identifies three new melanoma susceptibility loci. *Nat Genet.* 2011;43(11):1108–1113. doi:10.1038/ng.959, Gibbs et al, 2015, DOI: 10.1158/1055-9965) intronic snp...

In the earlier GWAS paper (Barrett et al. *Nat Genet* 2011), rs45430 was reported as a lead SNP that displays the lowest association P-value for melanoma risk in this locus. However, the

lowest P-value from a GWAS does not automatically qualify a SNP as a functional variant as they may just represent (or tag) the association signal of multiple highly correlated variants, including true functional variants.

A later GWAS meta-analysis in a larger population (Law *et al Nat Genet* 2015) in turn identified rs408825 as the lead SNP, which is in almost perfect LD ($r^2 = 0.99$ in EUR) with rs45430. In addition to these two SNPs, there are dozens of other SNPs, primarily located in the first intron of *MX2*, that are strongly correlated (**Figure 2A; Supplementary Table 9**), any of which could make strong candidates for follow-up analyses. To identify true functional variants from a set of highly correlated variants, they need to be functionally tested one by one, in addition to statistical deconvolution using fine-mapping approaches. Until now, a thorough functional assessment has not been done for rs45430 or other correlated SNPs in this locus, and our MPRA approach systematically tested all the qualified candidate variants including rs45430 and rs408825. Rather than these previously-reported lead SNPs, MPRA successfully identified rs398206 ($r^2 = 0.93$ with rs45430; $r^2 = 0.94$ with rs408825) as the melanoma-associated variant with the strongest evidence for functionality at this locus (**Supplementary Table 9**). We now include a more detailed explanation addressing this point in the **Discussion** section (first paragraph), including mentioning that MPRA includes an assessment of the lead SNP reported by Barrett and colleagues (2011).

-RNA seq experiments in *MX2* overexpressed vs control melanocytes (Figure6D) was performed after 72h of doxycycline treatment...this is too much time of induction- is of no surprise that this timings will not give much information on the downstream events. This experiment should be repeated with a shorter time point of induction 3-to 6h to narrow down the direct genes and possibly provide a mechanistic hypothesis.

Following the reviewer's suggestion, we performed RNAseq of *MX2* over-expressed in primary melanocytes at a time point with a shorter doxycycline treatment (**Supplementary Material**). For this analysis, we picked one of the three cell lines originally tested for RNAseq (C23, the line that displayed a growth effect with 100ng/ml doxycycline treatment over a 72hr period; **Fig 6B**). We first tested different durations of doxycycline induction (3, 6, 24, and 72 hr) on this melanocyte culture, and selected 6 hr as a time point for RNAseq based on level of *MX2* induction at the protein level (**Supplementary Fig 18D**). We compared transcriptomes of melanocytes with or without 100ng/ml doxycycline treatment at the 6hr time point and additionally assessed the 72hr time point as a control set in three biological replicates for RNAseq analysis. We did not include empty virus control at this time based on no apparent effect of doxycycline or virus transduction in enriched pathways (**Supplementary Table 16**).

Differentially expressed genes (DEG) in *MX2*-overexpressing melanocytes at both 6hr and 72hr were much fewer at FDR > 0.1 cutoff (compared to 158 DEG in the previous results). Given that we used only one cell line of a moderate level of induction for this analysis (6.2-fold at 6hr and 1.6-fold at 72hr at RNA levels; **Supplementary Table 15; Supplementary Fig 18E**), we relaxed the significance cutoff to select genes for pathways analyses. At nominal $P < 0.05$, 258 DEG were identified from 6hr induction (**Supplementary Table 17**) and 485 DEG from 72hr induction (**Supplementary Table 18**). Among the original 158 DEG, 132 genes (82%) displayed the same direction of fold change in the 72hr dataset from the second experiment, demonstrating consistency between two sets of experiments even with relatively lower detection levels in the second set. IPA analysis using 258 DEGs from 6hr induction displayed enrichment of pathways

including those involved in intracellular second messenger signaling, cancer, cell growth, neurotransmitters, cardiovascular signaling, and cellular immune response (**Supplementary Table 19**). A similar profile of pathways was commonly enriched in both 6hr and 72hr time points (**Supplementary Table 20**). Together with our initial finding, these results do not conclusively suggest a role of single dominant pathway, but rather multiple alternative hypotheses not excluding immune response-related mechanisms. The current results of earlier time point (6hr) detected much lower differential expression events compared to those from later time point (72hr) even with higher MX2 induction at mRNA and protein levels. These results suggest that short-term transcriptomic changes by MX2 induction in the melanocytes at 6hr are not greatly different from those reflected in the transcriptome at 72hr. Based on these results as well as the reviewer's comments, we conclude that we cannot rule out immune-related function of MX2 contributing to melanoma promotion in zebrafish, while it also remains possible that melanocyte-specific MX2 function is also involved. We have edited the text to make our conclusions clearer.

-The results that mostly support the role of MX2 in melanoma progression are definitely the zebrafish melanoma model, however this is the data that is least characterized and analyzed. And the in vitro opposite results further render a more thorough characterization of the MX2 melanoma zebrafish model.

Do not agree with the claims in 376...the in vitro data shows that MX2 reduces growth of melanocytes, but overexpressing MX2 in vivo increases the incidence of melanoma... Therefore, in my understanding, these results allude to a non-cell autonomous role of MX2...or otherwise please clarify better...

We agree with this comment, the conclusion in previous line 376 was poorly stated. Our conclusion was solely based on the MX2 effect *in vivo* as MX2 was expressed exclusively in melanocytic lineage in zebrafish. We fully acknowledge that we cannot rule out an effect, for example, through secreted factors from melanocytes that affects immune cells and or other cell types that could in turn affect tumor microenvironment, indeed, we feel this is a strong possibility for future study. We removed this conclusion about cell-autonomous effects.

Also the IL8 signalling pathway upregulation may indicate a role of innate immune cells/recruitment of neutrophils to the tumors...

This is a great point and definitely an interesting aspect for further study.

The authors should:

- Show representative images of the fish with melanoma to accompany the melanoma-free survival curves.

We now provide representative images of the fish in **Figure 6E**. A dozen of fish from each group were randomly photographed, and representative images are shown.

- Histopathology characterization of the MX2 melanoma: differentiation / infiltration /metastization /and pigmentation.

We agree that a thorough histopathological characterization is an important step to explore the molecular mechanism of MX2 function in melanoma promotion. However, these analyses could take up to 6 months or longer and should be left for future studies. Instead, we provide a qualitative analyses of tumor pigmentation in **Supplementary Fig 19A**. At median onset (when the most fish can be assessed), we observed no significant difference in the proportion of pigmented, mixed, or unpigmented tumors between MX2 and GFP control groups ($P = 0.27$, chi-square test comparing observed to expected proportions). Examples of qualitative assignment of each pigmentation category of tumors are shown in **Supplementary Fig 19B**.

- Quantification of number of events per fish along time – are there more foci – initiation events per animal?

We now provide the number of tumors per fish in **Supplementary Fig 19C**. The significant acceleration of tumor onset observed with MX2 overexpression (**Fig 6F**) suggests that more tumor-initiating events might occur in MX2-expressing fish compared to GFP-expressing fish. At median onset, we observed that >80% of fish with melanoma from both groups exhibited only one tumor. For the rest, they mainly had two tumors. Proportions of fish with one tumor compared to those with more than one tumors were not significantly different between MX2 and GFP groups ($P = 0.66$, chi-square test). However, these results may not fully rule out the possibility that MX2 overexpression initiates more events than GFP because tumors progress rapidly and may not allow enough time to witness the formation of a second tumor before fish have to be euthanized.

- Quantification of the size of the melanoma lesions (in vitro data show reduced growth – is it possible that there is an increase in the incidence but the melanomas are slow growers?)

We now provide size assessment for the melanoma lesions in **Supplementary Fig 19D**. In this analysis, we do see a significant departure of observed proportions of small, median, or large lesions between MX2 and GFP control groups, where MX2 group displayed a skew towards medium and large-sized lesions ($P = 0.028$, chi-square test). While this result shows larger size of tumors from MX2 group, it is hard to draw any conclusion beyond that the results reflect the early onset of melanoma in MX2 group demonstrated in the survival assay. An IHC to assess proliferation markers might be more informative to begin to answer the important question raised by the reviewer, but we believe it is beyond the scope of this manuscript. Examples of qualitative assignment of each size category of tumors are shown in **Supplementary Fig 19E**.

- Induce the MX2 melanoma model in innate immune deficient background (IRF8, Runx, Panther or Myd88 mutants, not all but 1 /2 if possible), to investigate a bit further the innate immune non-cell autonomous effect

This is an excellent point, and we could not agree more with the reviewer. This experiment, however, will take 8 months to 1 year and could be a subject for a future study.

MINOR

-Supplementary Figure 2 could be included in Figure 1...it would be most helpful for non experts ;)

We have now included this in **Figure 1A**.

Figure 6

-Panels of the figures do not appear in the same order of the text, becomes confusing. Starts in C- goes to A/B and then to D...

We have re-ordered the panels.

-line 301 should include “and MX2-low (bottom 25%; n = 28) melanocytes” from 106 individuals

We now edited the sentence as the reviewer suggested:

“To assess co-expressed genes and enriched pathways in melanocytes expressing *MX2* at a higher level, we profiled differentially expressed genes between *MX2*-high (top 25%; n = 28) and *MX2*-low (bottom 25%; n = 28) melanocytes **from 106 individuals**.”

Reviewers' comments:

Reviewer #1 (Remarks to the Author):

The authors have addressed all my concerns and should be published in the current version.

Reviewer #2 (Remarks to the Author):

Following the comments and suggestions of 3 reviewers, the authors have extensively revised their manuscript entitled „Massively parallel reporter assays combined with cell-type specific eQTL informed multiple melanoma loci and identified a pleiotropic function of HIV-1 restriction gene, MX2, in melanoma promotion“.

Reviewer #2 had indicated two major points and 19 minor points which all have now been addressed with great care and competence and to the satisfaction of the reviewer. The manuscript is complex and presents a large volume of data. This is clearly a challenge for the presenters but also a challenge for the readership. Nevertheless, the authors have done a good job and provide the plethora of information in a reasonable although quite expansive way. The specialized reader will be grateful for the detailed script. There are no further changes required.

Reviewer #3 (Remarks to the Author):

The authors addressed most comments, however a more detailed characterization of the MX2 melanoma fish melanoma is still missing.

I still believe that a histopathology characterization would be very interesting and paraffin sections with ki67 and histopathological analysis (degree of infiltration and morphometric analysis) should be no more than 2/3 weeks...unless the authors did not fix the fish...

Supl Fig 19

I could not find the number of fish analyzed – it is all in %. Please include the N.

We thank the reviewers for their constructive and helpful comments. We address reviewer comments below. Where changes have been made to the main text or supplement, we have highlighted such text in yellow.

Reviewers' comments:

Reviewer #1 (Remarks to the Author):

The authors have addressed all my concerns and should be published in the current version.

We thank the reviewer for the comment.

Reviewer #2 (Remarks to the Author):

Following the comments and suggestions of 3 reviewers, the authors have extensively revised their manuscript entitled „Massively parallel reporter assays combined with cell-type specific eQTL informed multiple melanoma loci and identified a pleiotropic function of HIV-1 restriction gene, MX2, in melanoma promotion”.

Reviewer #2 had indicated two major points and 19 minor points which all have now been addressed with great care and competence and to the satisfaction of the reviewer. The manuscript is complex and presents a large volume of data. This is clearly a challenge for the presenters but also a challenge for the readership. Nevertheless, the authors have done a good job and provide the plethora of information in a reasonable although quite expansive way. The specialized reader will be grateful for the detailed script. There are no further changes required.

We thank the reviewer for the comment.

Reviewer #3 (Remarks to the Author):

The authors addressed most comments, however a more detailed characterization of the MX2 melanoma fish melanoma is still missing.

I still believe that a histopathology characterization would be very interesting and paraffin sections with ki67 and histopathological analysis (degree of infiltration and morphometric analysis) should be no more than 2/3 weeks...unless the authors did not fix the fish...

We appreciate the reviewer's comment. We had not fixed the fish from the survival analyses for our initial submission, on which our estimation of time was based. During the first revision, we generated a small batch of additional fish for photographing, and we were able to fix some fish for future analyses. Using these fish samples, we performed histopathological analyses by H&E staining and immunohistochemistry with phosphorylated histone H3 (p-H3), a reliable marker for mitotic cells (Kim *et al. Oncotarget* 2017), that works better than Ki67 in our hands. Our data demonstrated that there is no apparent difference in tumor morphology and invasion between MX2 and GFP groups. We also did not observe significant difference in cell proliferation between two groups based on p-H3 staining. These data suggest that the effect of MX2 overexpression on tumor onset may not be due to increased proliferation but rather to earlier tumor initiation. We included these new data in **Supplementary Fig 20** and the main text, and provided individual raw images as a source data file, **SourceFile_SupFig20.zip**:

Results line 360-366

“We further performed histopathology analyses of zebrafish melanomas and did not observe gross difference in tumor morphology and invasion between MX2 and GFP groups (**Supplementary Fig 20A-D**). Immunohistochemistry for phospho-Histone H3 also did not display a significant difference in cell proliferation between two groups (P = 0.47, two-tailed t-test, n = 5; **Supplementary Fig 20E-G**). Together these data are consistent with *MX2* expression contributing to an increased melanoma risk, however consistent with *in vitro* data, likely through a mechanism other than increased cell proliferation.”

Discussion line 449-451

“Our zebrafish tumor proliferation assay also suggested that accelerated melanoma formation by MX2 might not be due to increased proliferation *per se*.”

Supl Fig 19

I could not find the number of fish analyzed – it is all in %. Please include the N.

We now added the number of fish for each group in **Supplementary Fig19** (MX2 = 88, GFP = 77).

REVIEWERS' COMMENTS:

Reviewer #3 (Remarks to the Author):

The authors have addressed all my concerns and should be published in the current version.